# Endosomal-lysosomal organellar assembly (ELYSA) structures coordinate lysosomal degradation systems through mammalian oocyte-to-embryo transition

Yuhkoh Satouh[1]*, Takaki Tatebe[1], Isei Tanida[2], Junji Yamaguchi[2], Yasuo Uchiyama[2], Ken Sato[1]*

[1]Laboratory of Molecular Traffic, Institute for Molecular and Cellular Regulation, Gunma University, Maebashi, Japan; [2]Department of Cellular and Molecular Neuropathology, Juntendo University Graduate School of Medicine, Tokyo, Japan

## eLife Assessment

This paper reports **important** findings on giant organelle complexes containing endosomes and lysosomes (termed endosomal-lysosomal organelles form assembly structures [ELYSAs]) present in mouse oocytes and 1- to 2-cell embryos. The data showing the localization and dynamics of ELYSAs during oocyte/embryo maturation are **convincing**. This work will be of interest to general cell biologists and developmental biologists.

*For correspondence:
yuhkohs@gunma-u.ac.jp (YS);
sato-ken@gunma-u.ac.jp (KS)

Competing interest: The authors declare that no competing interests exist.

**Abstract** Mouse oocytes undergo drastic changes in organellar composition and their activities during maturation from the germinal vesicle (GV) to metaphase II (MII) stage. After fertilization, the embryo degrades parts of the maternal components via lysosomal degradation systems, including autophagy and endocytosis, as zygotic gene expression begins during embryogenesis. Here, we demonstrate that endosomal-lysosomal organelles form large spherical assembly structures, termed endosomal-lysosomal organellar assemblies (ELYSAs), in mouse oocytes. ELYSAs are observed in GV oocytes, attaining sizes up to 7–8 µm in diameter in MII oocytes. ELYSAs comprise tubular-vesicular structures containing endosomes and lysosomes along with cytosolic components. Most ELYSAs are also positive for an autophagy regulator, LC3. These characteristics of ELYSA resemble those of ELVA (endolysosomal vesicular assemblies) identified independently. The signals of V1-subunit of vacuolar ATPase tends to be detected on the periphery of ELYSAs in MII oocytes. After fertilization, the localization of the V1-subunit on endosomes and lysosomes increase as ELYSAs gradually disassemble at the 2-cell stage, leading to further acidification of endosomal-lysosomal organelles. These findings suggest that the ELYSA/ELVA maintain endosomal-lysosomal activity in a static state in oocytes for timely activation during early development.

## Introduction

Fertilization triggers the dynamic remodeling of cellular components via zygotic gene expression as well as the degradation of cellular components from both maternal and paternal gametes (*Li et al., 2010*). After fertilization, maternal cytosolic proteins and RNAs are selectively degraded by the ubiquitin-proteasome system (UPS) (*Verlhac et al., 2010*) and specific RNA degradation systems (*Jiang and Fan, 2022*; *Schier, 2007*), respectively. Treatment with a proteasome inhibitor or knockdown of the zygote-specific proteasome assembly chaperone gene inhibits cytokinesis in zygotes,

**eLife digest** Mammalian egg cells, known as oocytes, store the nutrients and materials required for early embryo development. Before fertilisation, they exist in a low-energy, almost hibernation-like state. However, once fertilised, oocytes break down their stored nutrients to support embryo growth.

This process partly relies on the digestive activity of organelles called lysosomes, which contain proteins that can break down cell contents. Nutrients and unnecessary cell contents are delivered to lysosomes via compartments known as endosomes and autophagosomes, where they are then recycled to be used in development. However, it is important that these compartments become active at precisely the right time in development. If they are activated too soon, the stored nutrients will be depleted. On the other hand, if they are activated too late, the embryo will not have the nutrients it needs to develop.

To investigate how the oocyte controls activation of endosome and lysosome activity, Satouh et al. used live imaging techniques to observe mice oocytes as they matured, were fertilised and developed into early embryos. This revealed that early in oocyte development, a large spherical structure forms, which grows as the egg matures.

Naming the structure an Endosomal-Lysosomal Organellar Assembly (or ELYSA for short), Satouh et al. showed that before fertilisation, endosomes and lysosomes are grouped within ELYSAs, which keeps them mostly inactive. After fertilisation, the ELYSAs break apart, with normal endosomal and lysosomal activity beginning at the same time.

These findings shed light on an important, previously unknown regulatory process in oocyte maturation and early embryonic development. Recently, it has been suggested that endosomal activity may be involved in determining the developmental potential of an embryo. Therefore, studying assembly and disassembly of ELYSAs in ageing oocytes may provide insights into how fertility changes, with the potential to help improve fertility assessments and treatments in the future.

suggesting an essential role for the UPS in early development (*Shin et al., 2013*; *Shin et al., 2010*). Membrane components derived from each gamete are degraded by the lysosomal degradation system via autophagy and endocytosis (*Sato, 2022*; *Birgisdottir and Johansen, 2020*). In mice, autophagic activity is relatively low in metaphase II (MII) oocytes, whereas it is rapidly upregulated approximately 4 hr after fertilization, and LC3 (MAP1LC3: microtubule-associated protein light chain 3)-positive structures appear throughout the cytoplasm. Autophagic activity decreases slightly before and after the 2-cell stage but remains high from the 4- to 8-cell stage (*Tsukamoto et al., 2008*). Loss of Atg5 causes embryonic lethality at the 4- to 8-cell stages, suggesting that autophagy is essential for early mouse development. In contrast, some maternal plasma membrane (PM) proteins are internalized from the PM at the late 2-cell stage and are selectively degraded by lysosomes (*Morita et al., 2021*). Endocytosis inhibitors prevent cell division in 2-cell embryos, suggesting that endocytosis plays a pivotal role in early development (*Morita et al., 2021*; *Satouh and Sato, 2023*). Although lysosomal degradation systems are drastically upregulated after fertilization, their regulation remains unknown.

In *Caenorhabditis elegans*, the lysosomal activity in oocytes is upregulated by major sperm proteins secreted from sperms as the oocytes grow and mature (*Bohnert and Kenyon, 2017*). In this process, translational arrest of mRNA coding the V1-subunits of vacuolar ATPase in oocytes is released by the stimulation of sperm-derived factors. Consequently, the V1-subunits are recruited to the Vo-subunits in the lysosomal membrane, enhancing lysosomal acidification. This process is known as lysosomal switching (*Bohnert and Kenyon, 2017*). In mouse embryos, LysoTracker staining analysis suggests that lysosomal acidification is prominent at the 2-cell stage but not in MII oocytes and zygotes (*Tsukamoto et al., 2013*), raising the possibility that lysosomal switching takes place during mouse embryogenesis.

In this study, we aimed to investigate the dynamics of the endolysosomal system during mouse oocyte maturation and embryogenesis. We found that endosomes and lysosomes assembled to form spherical structures in germinal vesicle (GV)-stage oocytes, which became larger as they matured into MII oocytes. Ultrastructural analysis showed that these structures were large spherical membrane assemblies, including endosomes, lysosomes, and tubulo-vesicular structures located in the periphery of the PM. We refer to this structure as the endosomal-lysosomal organellar assembly (ELYSA). The

ELYSA contains several cytosolic constituents. While ELYSAs are disassembled in late-stage zygotes, immature endosomes and lysosomes gradually appear in the cytoplasm of early 2-cell stage embryos. Live imaging analysis showed that the signal of V1-subunit of V-ATPase on the acidic compartment became stronger during embryogenesis, suggesting further acidification of lysosomes. These findings suggest that the ELYSA maintains low endosomal and lysosomal activities and acts as a waiting place for endolysosomal organelles to undergo fertilization-triggered activation toward embryogenesis.

## Results

### Endosomes and lysosomes coordinately form giant structures in MII oocytes

To examine the localization of endosomes and lysosomes in mouse MII oocytes, early or late endosomes were stained with anti-RAB5 or -RAB7 antibody, respectively, and their localization was compared with that of lysosomes stained with anti-LAMP1 antibody. Giant structures harboring lysosomes with early and late endosomes were detected (*Figure 1A*, *Figure 1—figure supplement 1*, *Video 1*). Although small punctate structures that are RAB5-positive but LAMP1-negative also spread over the cytosol, most giant structures were positive for RAB5 and LAMP1 (*Video 1*). Each giant structure in the MII oocytes was spherical, ellipsoidal, or concatenated with small spheres. These structures were located in the peripheral region just beneath the PM, excluding the metaphase plate region in proximity to the metaphase chromosomes. These structures were observed from the GV stage to the pronuclear-zygote (PN-zygote) stage but were mostly disassembled at the 2-cell stage. Dispersion and separation of RAB5 and LAMP1 signals after the first division indicated that these giant structures were transiently formed (*Figure 1B*). On the other hand, RAB5-positive and LAMP1-negative punctate structures tend to accumulate along with LAMP1-positive punctate structures larger than 1 µm at the late 2-cell stage.

Analyses of the number, size, and distribution of LAMP1-positive organelles were carried out by reconstituting three-dimensional objects using LAMP1-positive signals in immunocytochemically stained oocytes and embryos (*Figure 2—figure supplement 1*). Although the LAMP1 signals on the PM increased in the 2-cell stages, reconstitution of the objects was carried out only for the cytoplasmic fraction. The total number of the LAMP1-positive objects was highest at the GV stage, and it decreased toward the late 2-cell stage, with a significant difference compared to that in MII oocytes (*Figure 2A*). Sorting of the objects based on their sizes (indicated with the diameters when converting the volume of each object to the corresponding sphere) revealed that the LAMP1-positive structures with diameters of 0.2–0.8 µm, which were prominent in GV oocytes, significantly decreased toward the MII stage, while the number of giant LAMP1-positive structures with diameters of 3–7 µm decreased during the post-fertilization changes at the early and late 2-cell stage relative to that in MII (*Figure 2B*). The ratio of the number of the LAMP1-positive structures clearly indicated a drastic increase in giant structures with 5–7 µm diameter during GV to MII transition (*Figure 2C*). The transiently increased LAMP1-positive structures with 9–10 µm diameter at the early 2-cell stage were observed as concatenated spheres that gathered upon cellular division; concurrently, prominent signals on the PM appeared, especially at the cleavage furrow and disappeared by the late 2-cell stage (*Figure 2D and E*). The total volume of LAMP1-positive objects was highest in MII, and it decreased over the early and late 2-cell stages (*Figure 2F*). LAMP1-positive structures with diameters of >3 µm were also observed in GV oocytes, but those larger than 5 µm in diameter were rarely detected, and the total volume of LAMP1-positive structures larger than 3 µm in diameter in MII oocytes accounted for 51.3% of the total volume. The large population of the LAMP1-positive structures with diameters larger than 3 µm in MII transitioned to structures with diameters of 1–3 µm during the 2-cell stages, as indicated by the percentage of the volume of each object size relative to the total volume (*Figure 2G*). The reconstitution of the objects above was conducted after the thresholding of the fluorescent intensity. On the other hand, objects with diameters of 6–7 µm in MII oocytes retained high intensity/volume, but those in GV oocytes or fertilized embryos revealed lower intensities (*Figure 2—figure supplement 2*).

Furthermore, the analysis of object distribution for GV and MII oocytes indicated that MII oocytes had a greater number of LAMP1-positive structures with >4 µm diameter in the cellular periphery, whereas GV oocytes had a greater number of these structures in the cellular medial regions (*Figure 2H*

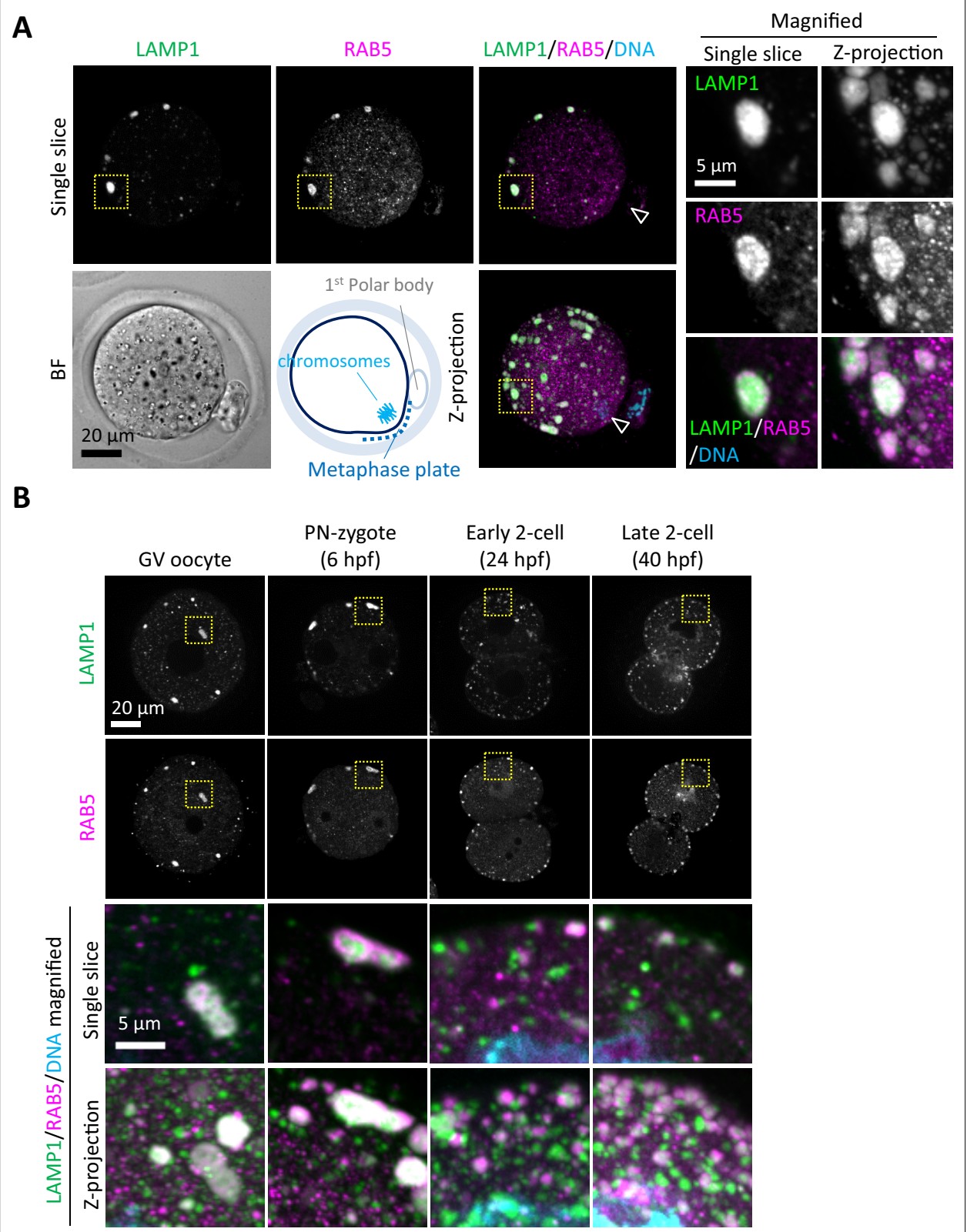

**Figure 1.** Endosomes and lysosomes form giant structures in the periphery of the oocyte plasma membrane. (**A**) Metaphase II (MII) oocytes were fixed and co-stained with anti-RAB5 and anti-LAMP1 antibodies. The schematic diagram indicates the location of oocyte chromosomes and metaphase plate in the MII oocytes. Arrowheads indicate the positions of oocyte chromosomes. Magnified regions (right) are indicated by yellow boxes. (**B**) Embryos at

*Figure 1 continued on next page*

*Figure 1 continued*
different stages were fixed and co-stained with anti-RAB5 and anti-LAMP1 antibodies. Hour(s) post-fertilization is indicated as hpf. DNA was stained with Hoechst 33342. Maximum intensity projection of confocal images at an axial scan range of 80 μm is shown as Z-projection images.

The online version of this article includes the following figure supplement(s) for figure 1:

**Figure supplement 1.** Endosomes and lysosomes form giant structures in the periphery of the oocyte plasma membrane.

*and I*). These results suggest that the giant LAMP1-positive structures are assembled and localized to the cellular periphery during oocyte maturation.

## Endosomes, lysosomes, and tubulo-vesicular structures form an ELYSA

To clarify the internal structure of this giant structure containing lysosomes and endosomes in the MII oocytes, we observed these in MII oocytes with in-resin correlative light and electron microscopy (CLEM) using immunological reaction (immune in-resin CLEM) with anti-RAB7 and -LAMP1 antibodies (*Tanida et al., 2023*; *Mitsui et al., 2023*). RAB7 and LAMP1 signals were well detected even in the ultrathin section (100 nm thickness) of the Epon-embedded MII oocyte, fixed with the mixture of 4% paraformaldehyde (PFA) and 0.025% glutaraldehyde (GA), which showed similar distribution as that shown in *Figure 1*, *Figure 3—figure supplement 1*. Toluidine blue staining in adjacent ultrathin sections revealed that the RAB7/LAMP1-double positive structures in the fluorescence observation were also positive for toluidine blue, with interspaces often found in the center of spheres (*Figure 3A*). Confocal fluorescent images of the ultrathin sections clarified that RAB7-positive signals were frequently observed on the periphery of these LAMP1-positive signals. Electron microscopic analysis of the RAB7/LAMP1-double positive region in the ultrathin sections revealed that the RAB7/LAMP1-double positive structures are assemblies of tubulo-vesicular structures, with either vacuolated or filled interiors (*Figure 3B*). The toluidine blue stain-positive structures were confirmed to match in size and peripheral distribution with those identified via RAB7/LAMP1 immunostaining, except for the metaphase plate, even under 2.5% GA conditions for precise ultrastructural observation (*Figure 3C*, *Figure 3—figure supplement 2*, *Video 2*). Therefore, we examined the ultrastructural morphology of toluidine blue-positive giant structures in MII oocytes fixed with 2.5% GA. The results showed that tubular membrane vesicles with a short diameter of approximately 60.5±15.9 nm assembled in the structure, with gaps in the cytoplasm of approximately 24.0±6.7 nm. Some of the vesicles also harbored lysosome-like electron-dense contents, while others displayed multilamellar or multivesicular body-like structures, and mitochondria appeared to be absent (*Figure 3D*). Based on these observations, we termed this membrane assembly the ELYSA.

## An autophagy regulator, LC3, also accumulated in ELYSA

We next examined whether autophagy-related proteins are included in ELYSAs. LC3 is a key regulator of autophagy and often used as an autophagosome marker. It has been suggested that the autophagic activity is decreased in MII oocytes since puncta formation of a transgenically expressed GFP fusion protein with LC3 disappeared at the MII stage (*Tsukamoto et al., 2008*). Since GFP fluorescence may be quenched under acidic conditions, we examined the localization of LC3 in GV and MII oocytes by immunostaining with an anti-LC3 antibody. We found that LC3 hardly formed puncta in the cytoplasm, but part of the LC3 signal overlapped with ELYSA in both GV and MII oocytes (*Figure 4A and B*), suggesting that LC3-positive membrane structures were also present in the ELYSA.

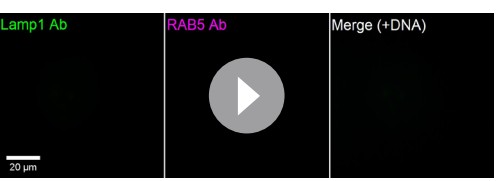

**Video 1.** Metaphase II (MII) oocytes were fixed and co-stained with anti-RAB5 and anti-LAMP1 antibodies. A series of the confocal images with a height of 80 μm is shown from top to bottom.

https://elifesciences.org/articles/99358/figures#video1

Further analysis of such double-positive structures for the LC3 and LAMP1 antibodies indicated that their number decreased drastically, while their total volume did not change during oocyte maturation (*Figure 4—figure supplement 1A and B*). Analysis of double-positive structures indicated that their average diameter significantly shifted from 2.36±0.31 μm (GV) to 3.78±0.20 μm (MII). Moreover, LAMP1-positive structures smaller than 2 μm in diameter were rarely positive for LC3 (*Figure 4—figure supplement 1C and D*). On the

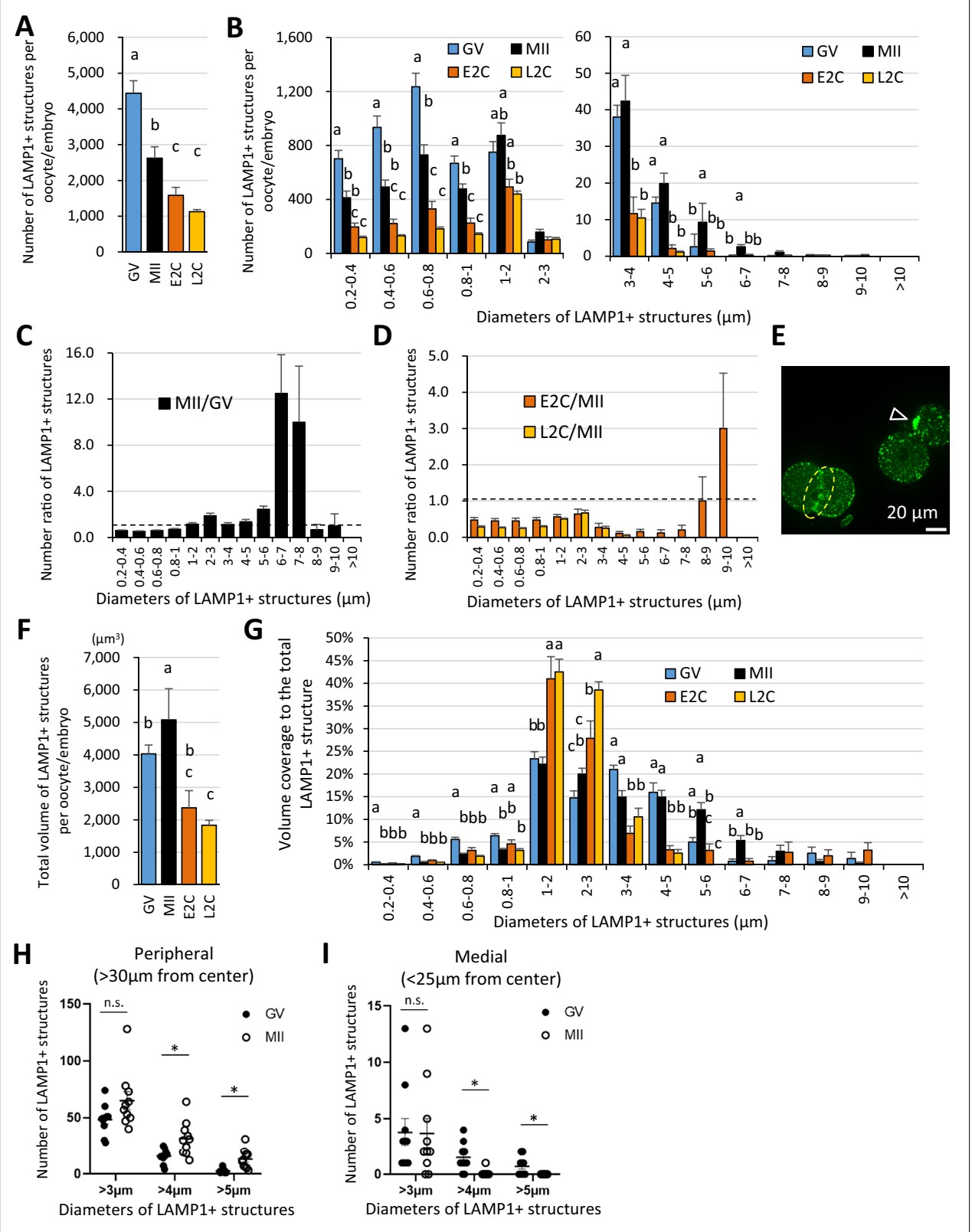

**Figure 2.** The giant structures are enlarged in the periphery of the metaphase II (MII) oocyte plasma membrane (PM) during oocyte maturation. Germinal vesicle (GV) oocyte, MII oocyte, early 2-cell (E2C: 24 hr post-fertilization [hpf]), and late 2-cell (L2C: 40 hpf) embryos were fixed and stained with anti-LAMP1 antibody. Reconstituted three-dimensional objects for the LAMP1-positive organelles in 9–10 oocytes for each stage from more than three independent experiments were further analyzed for number, size, and distribution. (**A**) Averaged total number of LAMP1-positive objects per

*Figure 2 continued on next page*

*Figure 2 continued*

oocyte/embryo. (**B**) Numbers of the LAMP1-positive objects sorted by size (indicated by the diameters of spheres calculated from volumes). (**C**) Ratios of (**B**) for MII/GV to present changes during oocyte maturation. (**D**) Ratios of (**B**) for E2C/MII or L2C/MII to present post-fertilization changes. (**E**) A single confocal section image for E2C embryos, which have transient and concatenated objects upon cellular division (arrowhead) and a prominent signal on the PM of cleavage furrow (dotted circle). (**F**) Averaged total volume ($\mu m^3$) of the objects per oocyte/embryo. (**F, G**) Total volumes of the objects sorted by size, or the percentage of each fraction relative to the total LAMP1-positive object volume. (**H, I**) Numbers of large objects in GV or MII oocytes at a distance of >30 μm (cellular peripheral) or <25 μm (medial) from the cellular center. Groups with different letters are significantly different (p<0.05, one-way ANOVA and Tukey's multiple comparison test). Error bars indicate SEM. Dot plots show mean ± SEM and asterisks indicate significant difference between two groups.

The online version of this article includes the following source data and figure supplement(s) for figure 2:

**Source data 1.** Excel file providing the numerical source data to *Figure 2*.

**Figure supplement 1.** Schemes of the three-dimensional (3D) particle analysis in this study.

**Figure supplement 2.** Large LAMP1-positive structures exhibit high fluorescent intensity, especially in metaphase II (MII) oocytes.

**Figure supplement 2—source data 1.** Excel file providing the numerical source data to *Figure 2—figure supplement 2*.

other hand, analysis of the distance from the cellular geometric center indicated that they tended to be observed at the periphery of oocytes in both stages (more than 80% in >30 μm in the MII oocyte), and no clear tendency of double positivity was observed along the distance (*Figure 4—figure supplement 1E and F*). A similar trend was observed for RAB5- and LAMP1-double positive structures, while the RAB5 signals tended to overlap with LAMP1 signals more specifically at the cellular periphery than LC3, even in the GV stage (*Figure 4—figure supplement 1G and H*).

## ELYSA enlargement proceeds in an actin-dependent manner

The localization of LAMP1-positive structures at the perinuclear (medial) region before oocyte maturation and the specific localization at the cell periphery after maturation (*Figure 2H and I*) is similar to the redistribution of the endoplasmic reticulum (ER) and the formation of cortical ER clusters. Fitzharris et al. showed that the inhibition of actin cytoskeleton polymerization disturbs ER migration and cortical ER cluster formation (*FitzHarris et al., 2007*). Thus, we examined the involvement of cytoskeletons in ELYSA formation by adding latrunculin A (LatA) or cytochalasin B (CCB) as inhibitors of actin polymerization and nocodazole (Noco) as an inhibitor of microtubule formation during in vitro maturation (IVM). While all the drugs inhibited polar body extrusion and metaphase plate formation as reported previously (*FitzHarris et al., 2007*), co-immunostaining of LC3 or RAB5 antibody with LAMP1 antibody revealed their persistent colocalization in ELYSAs (*Figure 5A*, *Figure 5—figure supplement 1*).

3D object analysis of the oocytes with or without cytoskeletal inhibitor treatment indicated that the total number of LAMP1-positive structures was increased only by CCB, and the total volume of the structures was unaffected (*Figure 5—figure supplement 2*). We then classified LAMP1-positive structures by their diameters and found that the number of structures with diameters of 0.8–2.0 μm increased when oocytes were treated with LatA and CCB; very few structures with diameters of 4–7 μm that corresponded to the enlarged ELYSAs were detected upon treatment with actin polymerization inhibitors, not the tubulin inhibitor (*Figure 5B*). Further examination of the distance from the cell center of each LAMP1-positive structure with a diameter larger than 4 μm in IVM oocytes indicated that only the actin polymerization inhibitors reduced the number of large structures at the cell periphery but not in the cellular medial region (*Figure 5C and D*).

## V1 component of V-ATPase partially localizes to the periphery of ELYSAs in oocytes

We studied the relationship between ELYSA formation and endosomal/lysosomal acidification in oocytes. V1A is a component of the V1-subunit of V-ATPase, which is associated with the outer lysosomal membrane and drives endosomal/lysosomal acidification. We used LAMP1-EGFP mRNA to observe ELYSA behaviors during oocyte maturation, with ATP6V1A-mCherry mRNA (V1A-mCherry: a core component of V-ATPase fused with the mCherry marker) to further examine the V1-subunit localization on ELYSAs. The mRNAs were injected into GV oocytes, and the oocytes were applied to IVM.

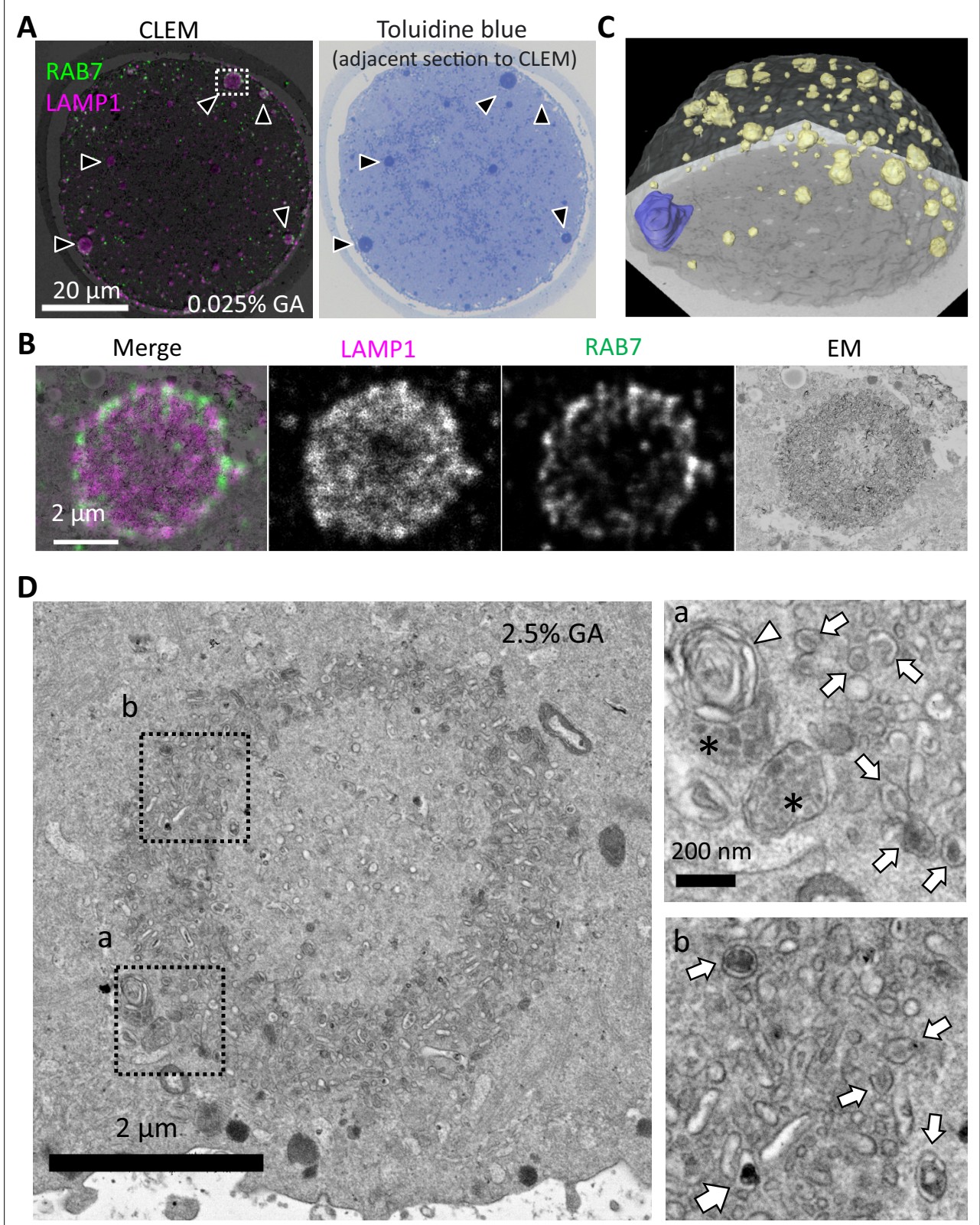

**Figure 3.** Endosomal-lysosomal organelles form assembly structures, ELYSA. (**A**) The internal structure of the giant structure in metaphase II (MII) oocytes was analyzed by correlative light and electron microscopy (CLEM) analysis using anti-RAB7 and anti-LAMP1 antibodies. A merged image of immunostaining and electron microscopy (EM) is indicated (left), and a toluidine blue staining image of the adjacent thin section (right) is indicated. Black arrowheads indicate the giant structures positive for RAB7/LAMP1/toluidine blue staining. Note that RAB7/LAMP1 staining is indicated in a

*Figure 3 continued on next page*

*Figure 3 continued*

pseudo-color. (**B**) The fluorescent distribution of RAB7/LAMP1 immunostaining was analyzed with deconvolution confocal microscopy. Note that RAB7/LAMP1 staining is indicated in pseudo-color. (**C**) A series of images of toluidine blue-stained serial semi-thin sections of the Epon-embedded MII oocyte were reconstructed in three dimensions (3D). Based on the images obtained, chromosomes were segmented blue, toluidine blue-positive structures were segmented yellow, and the oocyte plasma membrane (PM) was segmented white. (**D**) The internal structures of the toluidine blue-positive structure, ELYSA, were observed using conventional EM of oocytes fixed with 2.5% glutaraldehyde (GA). Magnified regions in a or b (right) are indicated by boxes (left). Arrows, arrowheads, or asterisks indicate vesicles harboring high electron density contents and multilamellar or multivesicular structures, respectively. ELYSA, endosomal-lysosomal organellar assembly.

The online version of this article includes the following figure supplement(s) for figure 3:

**Figure supplement 1.** Giant structures in metaphase II (MII) oocytes were immunostained with anti-RAB7 and -LAMP1 antibodies after glutaraldehyde (GA) fixation.

**Figure supplement 2.** Toluidine blue-positive structures in metaphase II (MII) oocytes show similar localization patterns to the endosomal-lysosomal organellar assembly (ELYSA).

3D object analysis for the matured oocytes at the MII stage expressing LAMP1-EGFP revealed a larger number of smaller LAMP1-positive structures (diameter of 0.2–0.4 μm) and a smaller number of structures with diameters of 0.6–1.0 μm, compared to immunostaining (*Figure 2*), but not significant difference in larger size (*Figure 6—figure supplement 1*). Live-cell imaging of LAMP1-EGFP fluorescence captured the dynamics of LAMP1-positive organelles during oocyte maturation, notably demonstrating its stable expression after GV breakdown (GVBD, 2–3 hr after initiation of IVM), and revealed that the enlargement of ELYSAs larger than 5 μm in diameter occurred through the sequential assembly of smaller ELYSAs that constantly migrated and contacted each other (*Figure 6A*, *Video 3*).

We then analyzed the localization of V1A-mCherry in the oocytes after IVM. Although it was indistinguishable whether V1-subunit molecules or V1-subunit-positive compartment were recruited to ELYSAs, V1A-mCherry signals were largely distributed on the ELYSA surface and rarely observed within ELYSAs in deconvolved confocal images (*Figure 6B*, *Video 3*).

## Acidification of lysosomes is facilitated as ELYSA disassembly proceeds in embryos

We examined the relationship between ELYSA disappearance and the emergence of acidic membrane compartments during early development. LysoSensor probes exhibit increased fluorescence in a pH-dependent manner upon acidification and enable the semiquantitative analysis of cellular acidic compartments, whereas LysoTracker probes exhibit non-pH-dependent fluorescence. We first co-stained oocytes and embryos at various stages with LysoTracker Red and LysoSensor Green (in the same medium drop). LysoTracker showed a pronounced puncta-like signal after the 2-cell stage, as previously reported (*Tsukamoto et al., 2013*). In contrast, the ELYSAs in GV and MII oocytes were clearly identified by LysoSensor, whereas LysoTracker only produced a very dim signal. Thus, the ratio of LysoTracker/LysoSensor fluorescent intensities in the cytoplasm of each stage embryo indicated that ELYSAs were present at a low ratio. In contrast, embryos at the 4-cell stage and beyond displayed small punctate signals with a higher ratio (*Figure 7*). A comparison between LysoTracker and V1A-mCherry signals showed that these fluorescent signals colocalized well on small bright vesicles but not on the ELYSA. Both LysoTracker and V1A-mCherry puncta were reduced in embryos cultured in the presence of bafilomycin A₁, which inhibits the H⁺

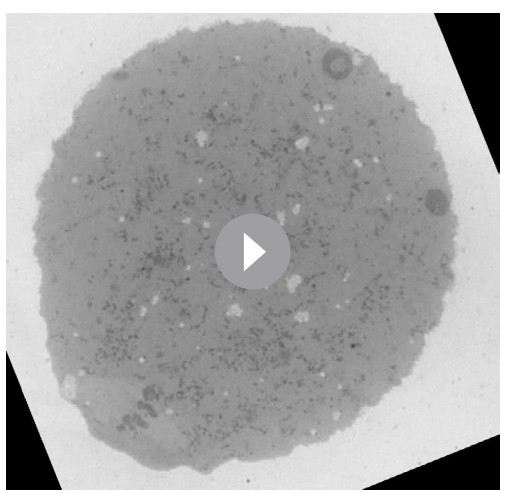

**Video 2.** A series of continuous toluidine blue-stained light microscopic images of the same oocyte (top half) reconstructed to three dimensions (3D). Chromosomes in the images obtained were segmented blue, toluidine blue-positive structures were segmented yellow, and oocyte plasma membrane (PM) was segmented white.
https://elifesciences.org/articles/99358/figures#video2

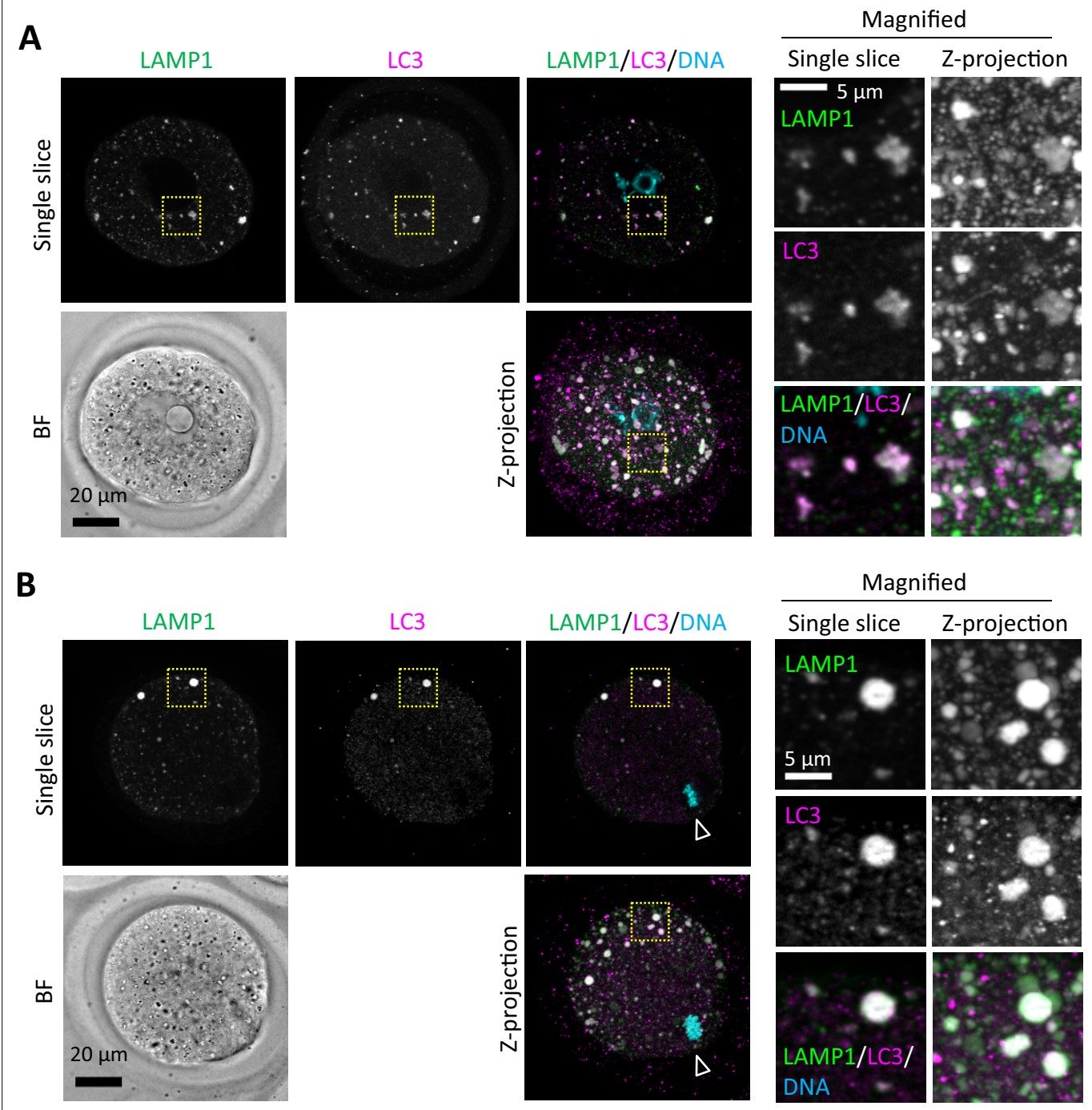

**Figure 4.** An autophagy regulator LC3 is detected in endosomal-lysosomal organellar assembly (ELYSA). Germinal vesicle (GV) (**A**) and metaphase II (MII) (**B**) oocytes were fixed and co-stained with anti-LC3 and anti-LAMP1 antibodies. Magnified regions (right) are indicated by yellow boxes. DNA was stained with Hoechst 33342. Arrowheads indicate the positions of oocyte chromosomes. Maximum intensity projection of confocal images at an axial scan range of 80 µm is shown as Z-projection images.

The online version of this article includes the following source data and figure supplement(s) for figure 4:

**Figure supplement 1.** LC3 signals colocalize with LAMP1 signals in a size-dependent manner.

**Figure supplement 1—source data 1.** Excel file providing the numerical source data to *Figure 4—figure supplement 1*.

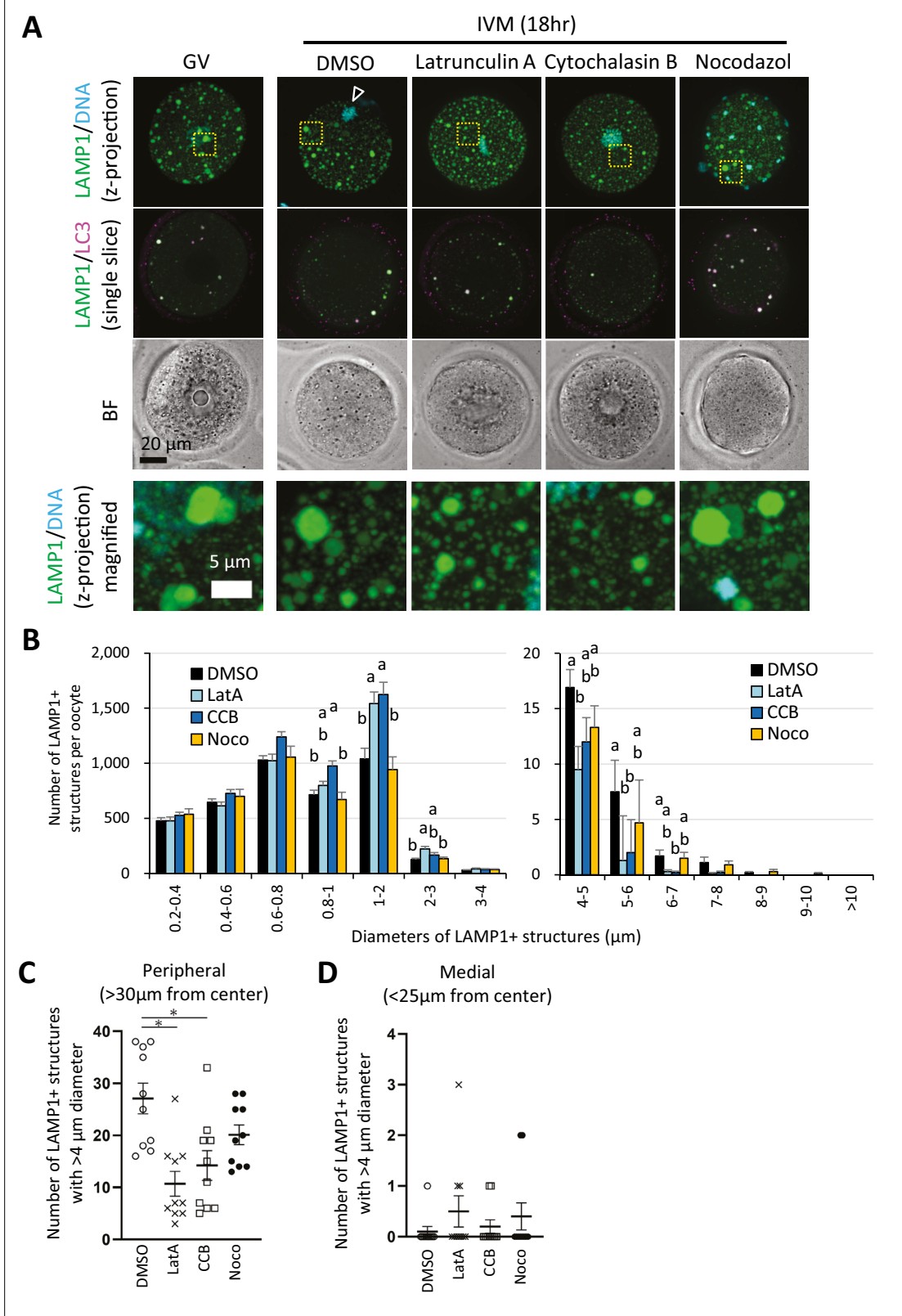

**Figure 5.** Enlargement of endosomal-lysosomal organellar assembly (ELYSA) and its redistribution to the cell periphery occur in an actin cytoskeleton-dependent manner. Germinal vesicle (GV) oocytes were incubated for in vitro maturation (IVM) for 18 hr with or without actin polymerization inhibitors (10 μM latrunculin A or cytochalasin B) or a tubulin inhibitor (20 μM nocodazole), then fixed and co-stained with anti-LC3 and anti-LAMP1 antibodies. The reconstituted objects for LAMP1-positive organelles in 10 oocytes for each treatment from more than three independent experiments were

*Figure 5 continued on next page*

*Figure 5 continued*

analyzed for the number, size, and distribution. (**A**) Maximum intensity projection of confocal images at an axial scan range of 80 µm is shown as Z-projection images. DNA was stained with Hoechst 33342. Arrowheads indicate the metaphase plate. (**B**) Average total LAMP1-positive object number per oocyte after sorting the objects by size (indicated by the diameters of spheres calculated from volumes). (**C, D**) Numbers of objects with diameters>4 µm in oocytes at a distance of >30 µm or <25 µm from the cellular center. Groups with different letters are significantly different (p<0.05, one-way ANOVA and Tukey's multiple comparison test). Error bars indicate SEM. Dot plots are indicated with mean ± SEM and asterisks indicate significant difference between two groups.

The online version of this article includes the following video, source data, and figure supplement(s) for figure 5:

**Source data 1.** Excel file providing the numerical source data to *Figure 5* and *Figure 5—figure supplement 2*.

**Figure supplement 1.** An early endosome marker RAB5 is localized within the endosomal-lysosomal organellar assembly (ELYSA) in the presence of inhibitors for cytoskeletons.

**Figure supplement 2.** Treatment with inhibitors for cytoskeletons affected the number of LAMP1-positive objects but not their volume.

**Figure 5—video 1.** Germinal vesicle (GV) oocytes were injected with the mixture mRNA solution for Lamp1-EGFP and V1A-mCherry, and incubated for in vitro maturation (IVM).

https://elifesciences.org/articles/99358/figures#fig5video1

---

translocation of V-ATPase (*Figure 7—figure supplement 1*). This suggests that LysoTracker detects membrane compartment acidification. However, it may not sufficiently capture this process within the ELYSA. In contrast, the results using LysoSensor suggest that lysosomes within the ELYSA were not completely neutralized and remained acidic at a lower level, even though V1A was difficult to localize.

When LAMP1-EGFP mRNA was injected into fertilized oocytes, it was difficult to visualize the lysosomes due to the presence of a population that migrates onto the PM during the 2-cell stage. Thus, to examine the spatiotemporal relationship between V1A localization and lysosomal acidification during embryogenesis, we injected V1A-mCherry mRNA into PN-zygotes and cultured them in the presence of LysoSensor. It should be noted that stable detection of LysoSensor and V1A-mCherry fluorescence required 4–8 hr. In this process, localization of V1A signal to lysosomes was rarely observed until ELYSA disappearance or disassembly throughout the PN-zygote stage, but rapidly enhanced in the early 2-cell embryo stage. LysoSensor-positive punctate structures, where V1A-mCherry signal colocalize, increased in number during the transition from early to late 2-cell stage, indicating that lysosomal acidification is promoted after ELYSA disappearance (*Figure 8*, *Video 4*).

## Low-level cathepsin-dependent proteolytic activity was maintained in ELYSA

Cathepsins are mainly responsible for the proteolytic activity in lysosomes. We visualized the proteolytic activity of cathepsin B, the expression of which has been confirmed in mouse embryos (*Tsukamoto et al., 2013*). Mouse oocytes and embryos at various stages were simultaneously collected and stained with Magic Red Cathepsin B staining solution to examine their relationship with the ELYSA. Magic Red staining showed that proteolytic activity was detected not only in the small, isolated membrane structures (mature lysosomes) but also in the ELYSA in the GV and MII oocytes, whereas small punctate structures were predominantly stained in other stages (*Figure 9A*, *Figure 9—figure supplement 1A*). The intensity plots of LysoSensor and Magic Red fluorescence revealed that the proteolytic activity indicated by the Magic Red signal in the ELYSA was detected, but not as strong as that in isolated punctate structures (lysosomes) in the GV and MII oocytes, and in developing embryos (*Figure 9B*, *Figure 9—figure supplement 1B and C*).

## Discussion

In this study, we identified a large spherical structure, the ELYSA, consisting of endosomes, lysosomes, LC3-positive structures, and various vesicular-tubular structures in mouse oocytes. Small ELYSAs are present in GV oocytes and increase in size (3–7 µm in diameter) through assembly at the cell periphery in an actin-dependent manner during oocyte maturation and occupying more than half the volume of the LAMP1-positive fraction in MII oocytes. After fertilization, as ELYSAs gradually disassemble in zygotes, punctate structures positive for either RAB5 or LAMP1 signal alone appear in the cytoplasm of embryos at the early 2-cell stage. The number of membrane structures positive for V1A-mCherry

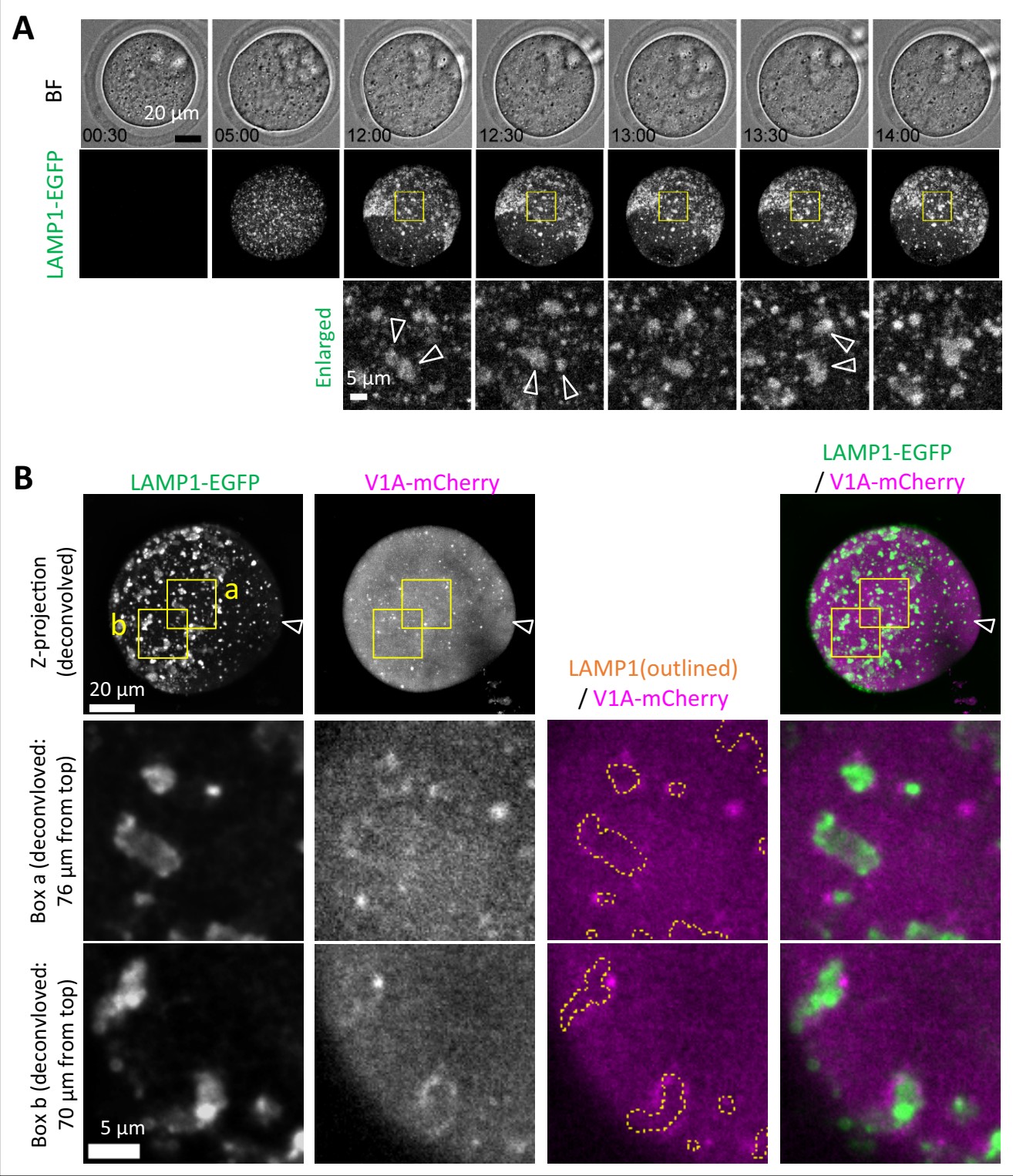

**Figure 6.** Assembly of endosomal-lysosomal organellar assembly (ELYSA) during migration and limited interaction with the acidification machinery. (**A**) Snapshots of a time-lapse observation for EGFP fluorescence of germinal vesicle (GV) oocytes injected with *Lamp1-EGFP* and *V1A-mCherry* mRNA are indicated. Maximum intensity projection of confocal images with a height of 40 μm (bottom half of the oocyte) are shown. Magnified perinuclear regions (right) are indicated by yellow boxes. Arrowheads indicate assemblies that adhere to each other in the subsequent frame. (**B**) Maximum intensity projection of deconvolved confocal images at an axial scan range of 80 μm is shown as Z-projection images. Magnified regions are indicated by yellow boxes (**a, b**) and the LAMP1-EGFP organelle regions are indicated by yellow dotted line.

*Figure 6 continued on next page*

*Figure 6 continued*

The online version of this article includes the following source data and figure supplement(s) for figure 6:

**Figure supplement 1.** Number and distribution of large LAMP1-EGFP structures do not significantly differ from intrinsic LAMP1 examined with immunofluorescence.

**Figure supplement 1—source data 1.** Excel file providing the numerical source data to *Figure 6—figure supplement 1*.

increases upon ELYSA disassembly, indicating further acidification of the endosomal/lysosomal compartment. These results suggest that the ELYSA functions as a reservoir that accumulates endosomal and lysosomal compartments at the periphery of the cell cortex and maintains their activities in a static state until embryogenesis following fertilization.

We studied the dynamics and activity of endosomes and lysosomes in mouse oocytes and embryos and found that some oocyte PM proteins are internalized from the PM at the late 2-cell stage and are selectively degraded in lysosomes (*Morita et al., 2021*). In this process, we noted that lysosomes were not well stained by LysoTracker until the early 2-cell stage, but their signals became stronger from the late 2-cell stage, suggesting the existence of a mechanism underlying developmentally regulated endosomal/lysosomal activation during embryogenesis (*Morita et al., 2021*; *Tsukamoto et al., 2013*). Notably, ELYSAs were stained with toluidine blue and LysoSensor, suggesting that these structures were acidified to some extent (*Figures 3 and 7*). In addition, Magic Red, which reflects cathepsin B activity, weakly stained ELYSAs (*Figure 9*). These observations indicate that lysosomes in ELYSAs have some activity, albeit at low levels. Mouse oocytes accumulate several maternal constituents, including proteins, lipids, and mRNA, and grow to approximately 75 µm in diameter. In addition, oocytes must remain at the GV and MII stages for an extended period to maintain their quality until fertilization. In such situations, lysosomal degradation activity may be suppressed to a relatively low level by the ELYSA, while nutrient uptake, storage, and protein synthesis are predominantly upregulated in growing oocytes. Furthermore, we found that the V1-subunit signals were largely detected on the periphery of ELYSAs but not in their interior in oocytes (*Figure 6*). Thus, it is possible that ELYSA formation prevents the localization of V1-subunits on the lysosomal membrane inside the ELYSA.

Ultrastructural analysis revealed that the ELYSA consists of outer and inner membrane assembly layers containing several endosomes and lysosomes, respectively (*Figures 1 and 3*). Considering the large number of RAB5-positive and LAMP1-negative punctate structures in MII oocytes, these layers may also reflect the assembly mechanism of the ELYSA. On the contrary, upon disassembly of the ELYSA, endosomes and lysosomes are expected to be released from the ELYSA earlier and later, respectively. This may explain the time lag between fertilization and lysosomal acidification beginning at the early 2-cell stage. Since the ELYSA appears to contain cytosolic constituents, it may accommodate some cytosolic constituents as a capsule to provide nutrition or signaling factors for development.

ELYSAs were distributed relatively widely in the GV oocyte, then specifically located at the periphery of the oocyte cell cortex, except in the MII cell plate in the MII stage (*Figures 1 and 2*, *Figure 4—figure supplement 1*). This change in distribution is reminiscent of that of the ER and cortical ER clusters, whose distribution is regulated by the actomyosin system (*FitzHarris et al., 2007*). In fact, the treatment with actin cytoskeleton polymerization inhibitors, but not a tubulin polymerization inhibitor, prevented the redistribution of ELYSAs from the cellular medial region to the cell periphery in oocytes (*Figure 5*). The former treatment increased the number of small LAMP1-positive organelles but reduced the size of ELYSAs in oocytes (*Figure 5*). These results suggest that actin cytoskeleton is involved in the assembly of smaller (<2 µm in diameter) LAMP1-positive organelles for the enlargement of ELYSAs in the cell periphery.

Our detailed imaging analysis showed that lysosomal acidification is promoted simultaneously with the detection of the V1-subunits on lysosomes during ELYSA disassembly in late zygotes and is further facilitated from the 2- to

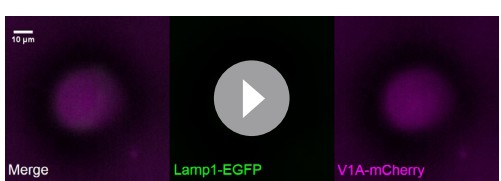

**Video 3.** Germinal vesicle (GV) oocytes were injected with V1A-mCherry mRNA and incubated for 18 hr for in vitro maturation (IVM) in the presence of LysoSensor green. Fluorescent images of a representative oocyte which matured to metaphase II (MII) stage are presented. A series of the confocal images of 81 slices with a height of 80 µm is shown from top to bottom.
https://elifesciences.org/articles/99358/figures#video3

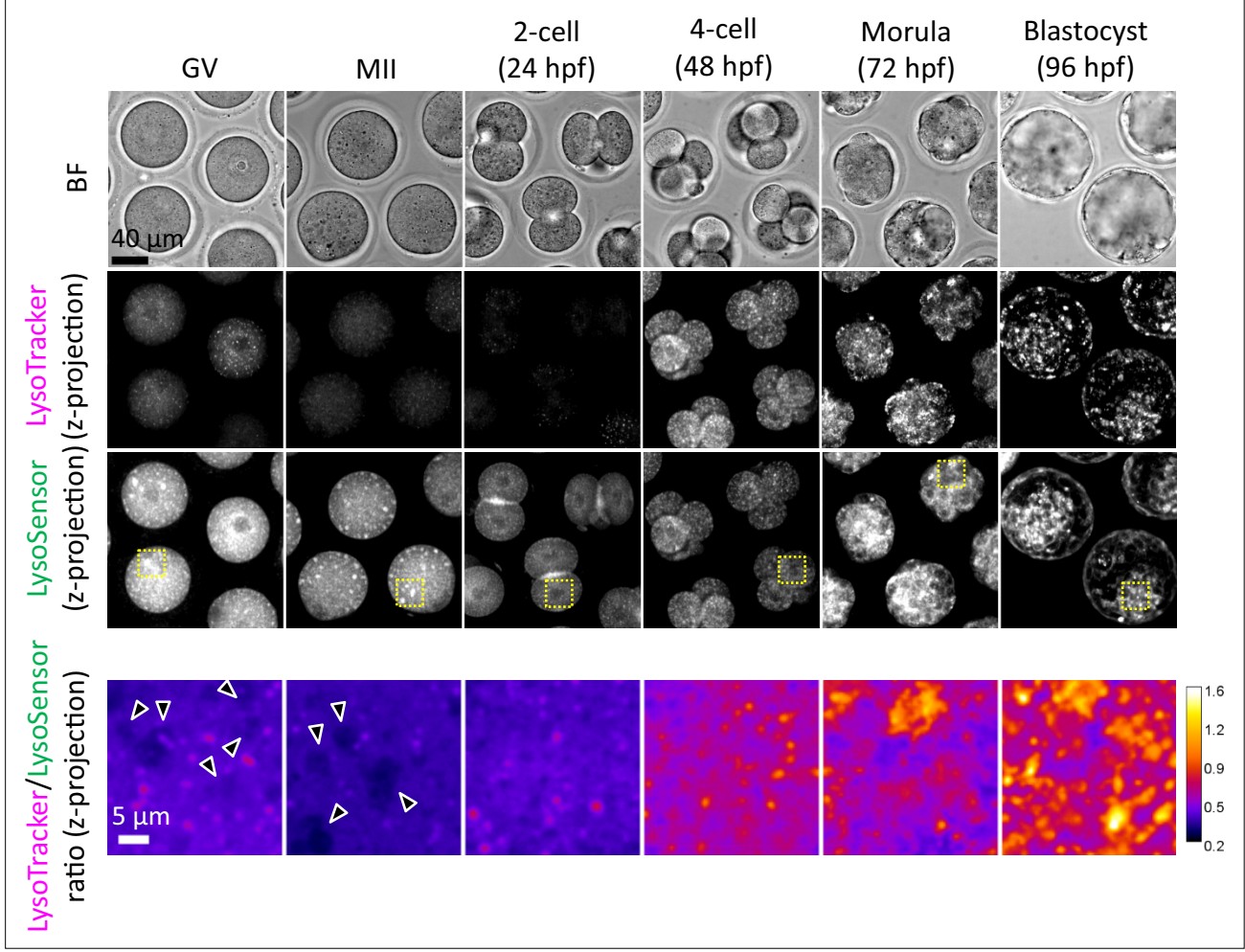

**Figure 7.** Endosomes and lysosomes in the endosomal-lysosomal organellar assembly (ELYSA) are weakly acidified, but further acidification occurs after the 4-cell stage. Oocytes and embryos at different stages were stained in the same drop for LysoSensor Green and LysoTracker Red staining analysis, and confocal images were acquired. Averaged intensity projection of confocal images at an axial scan range of 80 μm is shown as Z-projection images. Magnified ratiometric image for LysoTracker/LysoSensor fluorescence of cytosolic regions indicated by yellow boxes are presented as 'fire' lookup table images at the bottom. Arrowheads indicate ELYSAs recognized with LysoSensor.

The online version of this article includes the following figure supplement(s) for figure 7:

**Figure supplement 1.** LysoTracker-positive punctate structures colocalized with V1A-mCherry.

4-cell stage and is enhanced after the morula stage (*Figure 8*, *Figure 7—figure supplement 1*). These findings suggest that lysosomal acidification and maturation are linked to the ELYSA disassembly. Notably, minor and major zygotic gene activation in mouse embryos occur at the 1- and 2-cell stages, respectively, implying that lysosomal degradation of maternal components is coupled with zygotic gene expression to promote embryogenesis (*Schulz and Harrison, 2019*). Fertilization appears to trigger ELYSA disassembly, but the downstream signaling pathways remain to be identified and warrant further research.

Very recently, after sharing this ELYSA paper as a preprint in 2023 (*Satouh et al., 2023*), *Zaffagnini et al., 2024* reported the endolysosomal vesicular assemblies (ELVAs), which resemble ELYSAs, in mouse oocytes. The ELVA is defined as super-organelles containing endosomes, lysosomes, autophagosomes, and proteasomes along with cytosolic proteins, which sequesters detrimental protein aggregates inside and degrade them upon oocyte maturation (*Zaffagnini et al., 2024*). Consistencies between the ELYSA and the ELVA in their contents and actin-dependent dynamics during oocyte maturation/embryogenesis strongly suggest that ELYSAs and ELVAs are identical structures (*Zaffagnini et al., 2024*), assuring the solidity of these findings in this paper. Zaffagnini et al. also observed the

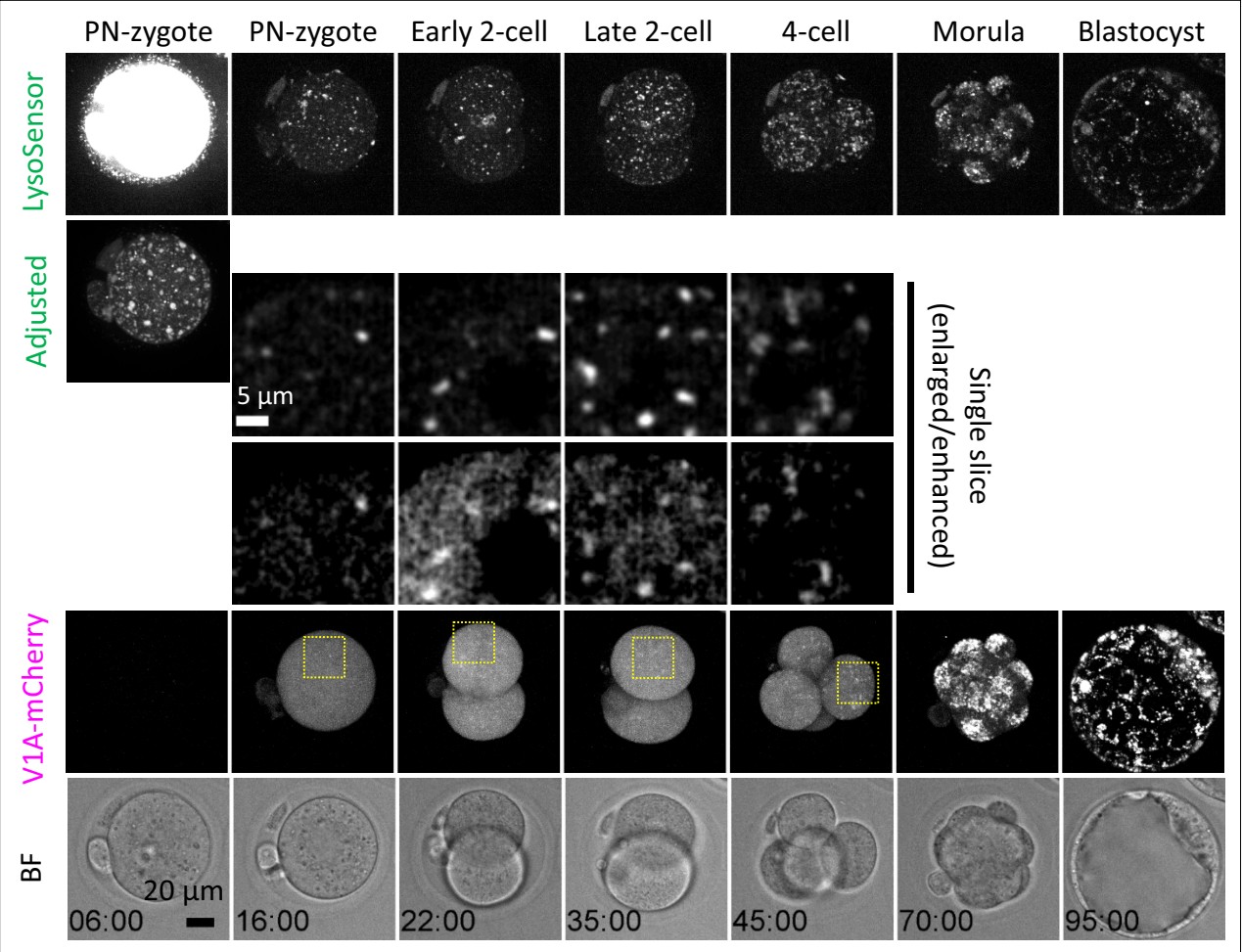

**Figure 8.** ATP6V1A localization on acidic compartments increases as endosomal-lysosomal organellar assemblies (ELYSAs) are disassembled. Embryos injected with *V1A-mCherry* mRNA (at 3 hr post-fertilization [hpf]) were incubated to the blastocyst stage in the presence of LysoSensor Green. Snapshots of a time-lapse observation of a representative embryo are shown as maximum intensity projection of confocal images with a height of 80 µm unless otherwise specified. Magnified perinuclear regions are indicated by yellow boxes.

accumulation of proteasomes and ubiquitinated proteins in ELVAs and demonstrated that the depletion of a RUN and FYVE domain-containing protein 1 (RUFY1) protein from oocytes by the Trim-Away method causes disappearance of ELVAs, suggesting that RUFY1 is involved in the ELVA assembly. Analysis of *Rufy1*-knockout ELVA/ELYSA-free oocytes is awaited to elucidate the physiological role of ELVAs or probably ELYSAs. They also suggested that ELVAs sequester and degrade specific proteins that are known to be aggregate and cause diseases, including proto-oncogene receptor tyrosine kinase KIT, upon oocyte maturation (*Zaffagnini et al., 2024*). Complementary to their findings, our study revealed that the recruitment of the V1-subunit is limited on ELYSAs in oocytes but enhanced on the isolated LAMP1-positive vesicles after late 2-cell stage. These findings support a hypothesis that ELYSA has a multimodal function, keeping endosomal and lysosomal compartments at a relatively static state and supplying these membrane compartments as a reservoir for embryogenesis.

Inhibition of lysosomal acidification by bafilomycin $A_1$ treatment arrests embryogenesis at the 4- to 8-cell stage, while inhibition of cathepsin activity using a mixture of E64d and pepstatin A blocks embryogenesis at the 8-cell or morula stage (*Tsukamoto et al., 2013*). In contrast, the inhibition of endocytosis by Pitstop2 treatment causes arrest at the 2-cell stage (*Morita et al., 2021*). These differences may imply that at the 2-cell stage, the endocytic activity is more important than the lysosomal degradation activity, which is essential at later stages. However, insights from this study shed light on the correlation between increased endocytic/autophagic activity and lysosomal maturation through

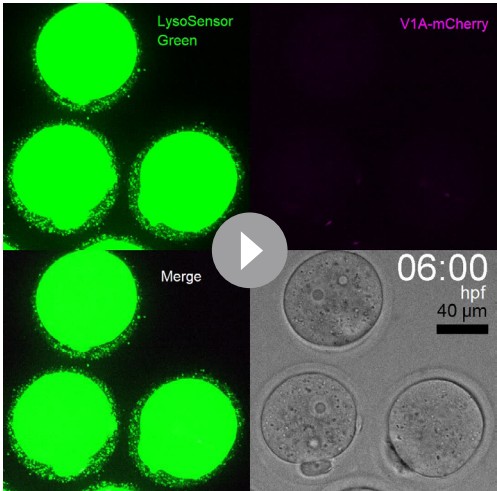

**Video 4.** Embryos were injected with V1A-mCherry mRNA at 3 hr post-fertilization (hpf) and incubated for further embryogenesis in the presence of LysoSensor Green during embryonic development. A time-lapse movie of a representative embryo is shown (maximum intensity projection of confocal images at a height of 80 µm).

https://elifesciences.org/articles/99358/figures#video4

ELYSA disassembly and the potential significance of minor lysosomal maturation in the early phase of embryogenesis.

The relationship between such endocytic/autophagic activities in early-stage embryos and the embryonic developmental potential has been recently studied. Using GFP-LC3 as an indicator of autophagic activity, Tsukamoto et al. found that aged oocytes from 14- to 15-month-old mice had lower autophagic activity than those from younger mice, probably due to decreased lysosomal activity. Females transplanted with embryos with high autophagic activity at the 4-cell stage have a significantly higher litter size ratio than that of control females (*Tsukamoto et al., 2014*). Analysis of ELYSA formation and disassembly in aged oocytes and embryos may be a useful tool for determining oocyte quality by examining appropriate regulation of endosomal/lysosomal activities.

## Methods
### Animal experiments and strains
Neither randomization nor blinding was used for animal selection (8–16 weeks of age). All experimental protocols involving animals were approved and performed in accordance with the guidelines of the Animal Care and Experimentation Committee of Gunma University (approval number 22-076). Hybrid B6D2F1 mice were purchased from the Japan SLC (Hamamatsu, Japan).

### Antibodies and inhibitors
The following primary antibodies were used: rabbit monoclonal anti-RAB5 (C8B1; 3547; Cell Signaling Technology, Danvers, MA, USA), rabbit monoclonal anti-RAB7 (D95F2; 9367; Cell Signaling Technology), rat monoclonal anti-mouse LAMP1 (1D4B; sc-19992; Santa Cruz Biotechnology, Dallas, TX, USA), and mouse monoclonal anti-rat LC3 (Clones 4E12; M152-3; Medical and Biological Laboratories, Tokyo, Japan). The following secondary antibodies were used: goat anti-rabbit IgG (H+L) conjugated with Alexa Fluor 555 (A-21428; Thermo Fisher Scientific, Waltham, MA, USA), goat anti-rat IgG (H+L) cross-adsorbed secondary antibody conjugated with Alexa Fluor 488 and 647 (A-11006 and A-21247, respectively; Thermo Fisher Scientific), and goat anti-mouse IgG (H+L) cross-adsorbed secondary antibody conjugated with Alexa Fluor 546 (A-11030; Thermo Fisher Scientific). The following inhibitors or solutions were used for IVM or embryonic culture assays: dimethyl sulfoxide (DMSO) (13406-55; Nacalai Tesque, Kyoto, Japan); LatA actin polymerization inhibitor that binds to monomer actin (428021; Sigma-Aldrich, St. Louis, MO, USA); CCB, an actin polymerization inhibitor that binds to the plus-end of polymerized actin (C6762; Sigma-Aldrich); and Noco, a tubulin polymerization inhibitor that binds to β-tubulin (M1404; Sigma-Aldrich). Stock solutions of LatA, CCB, or Noco were prepared by solubilizing each in DMSO at 200× concentration for the assays. Bafilomycin $A_1$ was dissolved in DMSO (L266; Dojindo, Kumamoto, Japan) at a final concentration of 25 nM.

### Oocyte preparation
GV oocytes were collected from the ovaries of female mice 46 hr after injection with pregnant mare serum gonadotropin or an anti-inhibin antibody (CARD HyperOva; Kyudo, Kumamoto, Japan) (*Takeo and Nakagata, 2015*). Antral follicles were suspended in 30 µL of FHM medium drops (Sigma-Aldrich) covered with liquid paraffin (Nacalai Tesque), punctured using 26G needles (Terumo, Tokyo, Japan), and the GV oocytes were collected using a glass needle with a mouth pipette. MII oocytes were collected from the oviducts of female mice 13 hr after injection with CARD HyperOva and 7.5 IU

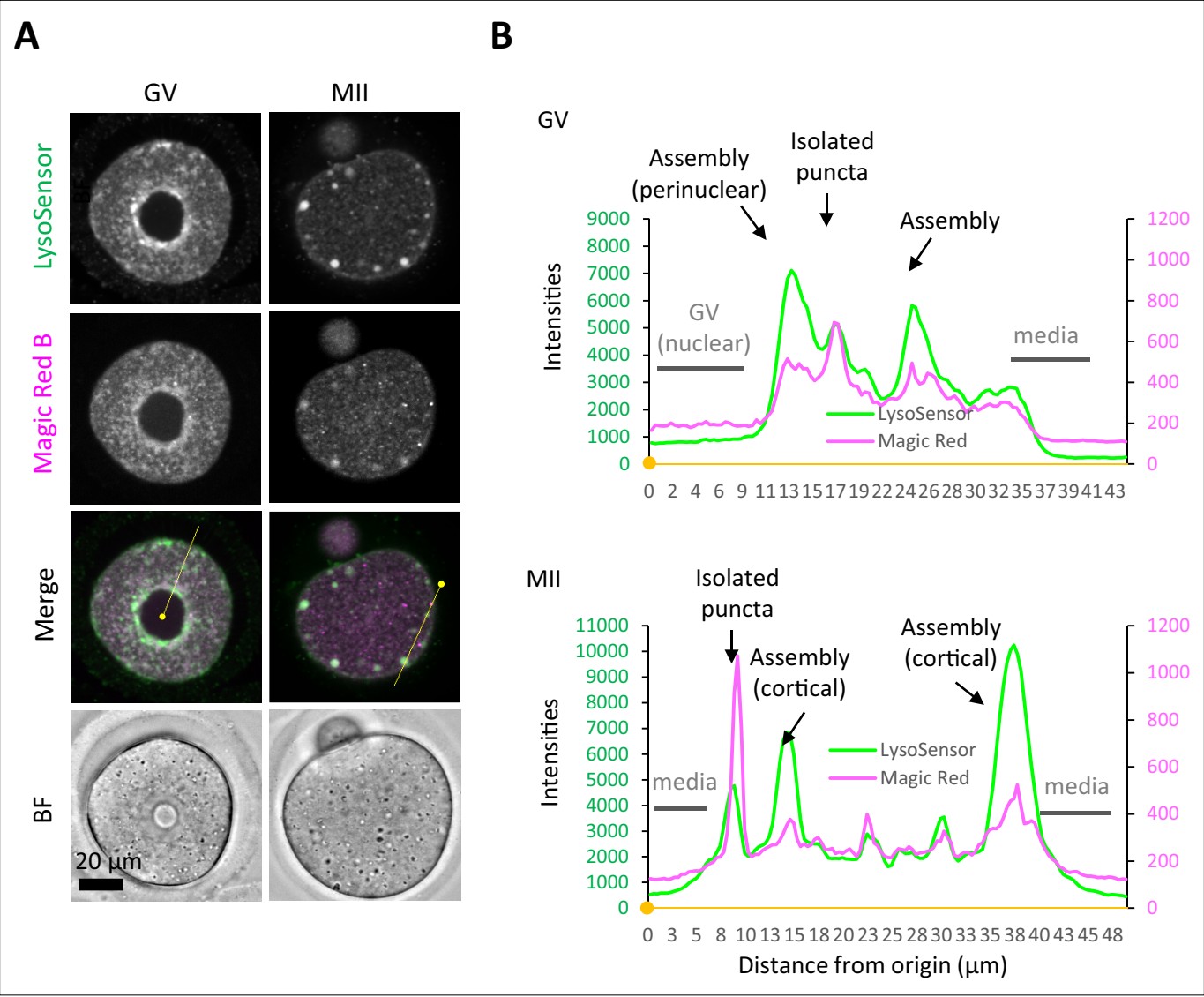

**Figure 9.** Proteolytic activity of cathepsin B is maintained at a relatively low level in the endosomal-lysosomal organellar assembly (ELYSA). (**A**) Germinal vesicle (GV) and metaphase II (MII) oocytes were stained in the same drop for LysoSensor Green and Magic Red Cathepsin B staining analysis, and confocal images were acquired. (**B**) Fluorescent intensities of LysoSensor and Magic Red on the yellow lines indicated in (**A**) are indicated by a line plot. Small, filled dots indicate the origin (0 µm) in distance.

The online version of this article includes the following source data and figure supplement(s) for figure 9:

**Source data 1.** Excel file providing the numerical source data to *Figure 9* and *Figure 9—figure supplement 1*.

**Figure supplement 1.** Magic Red Cathepsin B staining in germinal vesicle (GV) and metaphase II (MII) oocytes showed higher cytoplasmic signals but similar intensity in isolated punctate structures to developing embryos.

human chorionic gonadotropin (hCG; ASKA Pharmaceutical, Tokyo, Japan) 48 hr apart. The oviductal wall was punctured using a 26G needle to collect cumulus-oocyte complexes in 20 µL potassium simplex optimization medium (KSOM; Kyudo) drops covered with liquid paraffin.

## DNA vector construction and mRNA synthesis

Complementary DNA for mouse *Lamp1* (accession number NM_001317353) or *Atp6v1a* (accession number NM_001358204) was amplified using PCR with High-fidelity Taq Polymerase, KOD FX Neo (Toyobo, Osaka, Japan), and a mouse ovary cDNA library as a template (Genostaff, Tokyo, Japan). The amplicon was then subcloned into the entry vector pDONR221 and transferred into the destination vectors pDEST-CMV-C-EGFP (*Agrotis et al., 2019*) (a gift from Robin Ketteler [Addgene

plasmid # 122844; http://n2t.net/addgene:122844; RRID:Addgene_122844]; for *Lamp1*) or pDest-mCherry-N1 (*Hong et al., 2010*) (a gift from Robin Shaw [Addgene plasmid # 31907; http://n2t.net/addgene:31907; RRID:Addgene_31907]; for *Atp6v1a*) using Gateway recombination cloning technology. The following primer combinations were used for cloning: 5'-GCGCACAAGTTTGTAC AAAAAAGCAGGCTGCCACCATGGCGGCCCCCGGCG-3' and 5'- GCGCACCACTTTGTACAAGA AAGCTGGGTTGATGGTCTGATAGCCGGCGTGACT –3' (for *Lamp1*), or 5'- GCGCACAAGTTTGTAC AAAAAAGCAGGCTGCCACCATGGATTTCTCCAAGCTACCC-3' and 5'-GCGCACCACTTTGTAC AAGAAAGCTGGGTTGTCTTCAAGGCTACGGAATG-3' (for *Atp6v1a*). The cDNA encoding *Lamp1-EGFP* or *Atp6v1a-mCherry* were amplified using the following primer combinations: 5'-CGCAAATG GGCGGTAGGCGTG-3' and 5'-TCCAGCAGGACCATGTGATCGC-3' (for *Lamp1-EGFP*), or 5'-CGCA AATGGGCGGTAGGCGTG-3' and 5'-TTGGTCACCTTCAGCTTGG-3' (for *Atp6v1a-mCherry*). RNA synthesis using each PCR amplicon as a template, poly-A tail addition, and mRNA purification were performed using an mMESSAGE mMACHINE T7 Ultra kit (Thermo Fisher Scientific) and MEGA-clear transcription clean-up kit (Thermo Fisher Scientific), according to the manufacturer's protocols. Synthesized mRNAs were dissolved in RNase-free demineralized water at a concentration of 400 ng/ μL and frozen in aliquots until use.

## IVM, fertilization, and embryonic culture

IVM for the GV oocytes was conducted by washing the oocytes twice and culturing in alpha-modified Eagle's minimum essential medium (Thermo Fisher Scientific) containing 5% (vol/vol) fetal bovine serum (FBS), 0.1 IU/mL follicle-stimulating hormone (MSD, Tokyo, Japan), 1.2 IU/mL of hCG (ASKA Pharmaceutical), and 4 ng/mL of epidermal growth factor (Thermo Fisher Scientific) for 17 hr at 37°C under a 5% $CO_2$ atmosphere (*Morohaku et al., 2016*). For in vitro fertilization, spermatozoa from a B6D2F1 epididymis were pre-incubated in a 100 μL modified human tubal fluid medium (Kyudo) drop covered with liquid paraffin and inseminated at a sperm concentration of $1.5×10^5$ per mL for 1.5 hr. Excess spermatozoa were washed out, and the oocytes were incubated in a 20 μL KSOM drop (Kyudo) for further development. IVM assays using LatA and CCB were performed by transferring oocytes to medium drops containing each inhibitor or DMSO 4 hr after the start of IVM (waiting for the completion of GVBD), as previously reported (*FitzHarris et al., 2007*). IVM media containing DMSO, LatA, or CCB was prepared by diluting the stock solution (1:400), and the final concentration of the inhibitors was 10 μM. The embryonic culture assay using bafilomycin $A_1$ was conducted by transferring the PN-zygote into a 20 μL drop of KSOM containing 25 nM bafilomycin $A_1$ covered with liquid paraffin at 6 hr post-fertilization (hpf). Embryos were transferred at 24, 48, and 72 hpf to freshly prepared KSOM containing bafilomycin $A_1$.

## Immunostaining

Immunostaining of oocytes or embryos was performed as previously described (*Morita et al., 2021*). Oocytes or embryos were cultured to the indicated stages and fixed in 4% PFA (Nacalai Tesque) in phosphate-buffered saline (PBS) containing 0.1 mg/mL polyvinyl alcohol (PVA; Sigma-Aldrich; PVA/PBS) for 15 min at 25–27°C. After fixation, the cells were permeabilized with PVA/PBS containing 0.1% Triton X-100 (Sigma-Aldrich) for 30 min at 25–27°C and washed three times with PVA/PBS. After blocking with 10% FBS (Thermo Fisher Scientific) in PBS for 1 hr at room temperature, the cells were incubated with the aforementioned primary antibodies (1:100 for all) overnight at 4°C. Next, the cells were washed three times in PVA/PBS and incubated with anti-mouse, rat, or rabbit IgG secondary antibodies coupled with Alexa Fluor 488, 555, or 564 (1:200 for all; Thermo Fisher Scientific) at 25–27°C for 2–4 hr. Thereafter, the cells were incubated with PVA/PBS containing 10 μg/mL Hoechst 33342 (Dojindo) for 10 min and washed with fresh PVA/PBS a few times. Subsequently, control and nonspecific staining were performed by processing oocytes and embryos in the absence of primary antibodies. The oocytes and embryos were imaged in PVA/PBS microdrops in 35 mm glass-bottomed dishes (MatTek Corporation, Ashland, MA, USA). For each position, confocal images at 81 *z*-axis planes with 1 μm increments were acquired on an IX-71 inverted microscope (Olympus Corporation, Tokyo, Japan) equipped with a CSU W-1 confocal scanner unit (Yokogawa Electric Corporation) using a 60× silicone immersion lens UPLSAPO60XS (Olympus Corporation).

## LysoTracker/LysoSensor staining of oocytes and generation of ratio images

Acidic organelles in the oocytes were visualized using LysoTracker Red DND-99 and LysoSensor Green DND-189 (Thermo Fisher Scientific) according to the manufacturer's instructions. Briefly, the oocytes or embryos at target stages for one experimental round were obtained simultaneously and incubated in a large drop of reaction mixture (KSOM containing 1/10,000 Magic Red stock solution dissolved in DMSO and 1/1000 LysoSensor Green DND-189 stock solution dissolved in DMSO) for 30 min at 37°C under 5% $CO_2$ in a humidified atmosphere in liquid paraffin in a 35 mm glass-bottomed dish (MatTek Corporation). Subsequently, confocal images at 41 z-axis planes with 2 μm increments were obtained using a CSU W-1 spinning disk confocal microscope with a 30× silicone immersion lens UPLSAPO30XS (Olympus Corporation). Acquisition conditions for fluorescence excited by 470 nm (LysoSensor) and 555 nm (LysoTracker) lasers were determined in preliminary experiments to be almost 1:1 intensities between the two wavelengths for cytosolic punctate structures in the 2-cell embryo. Further generation of ratio images for LysoTracker/LysoSensor fluorescence were carried out using Fiji software (*Schindelin et al., 2012*). Averaged intensity projection was carried out, determining background intensity of each field by subtracting the averaged intensities of four cellular blank regions from the whole image. Then, 32-bit ratio images for LysoTracker/LysoSensor fluorescence of cytosolic regions were generated by processing images with 'Math/Divide' function in Fiji, and 'fire' lookup table images were converted to RGB images.

## Magic Red/LysoSensor staining

Intracellular cathepsin activity was assayed in oocytes or embryos using Magic Red Cathepsin B or Cathepsin L detection kits (ImmunoChemistry Technologies LLC, Davies, CA, USA) according to the manufacturer's instructions. Briefly, the oocytes or embryos at target stages for one experimental round were obtained simultaneously and incubated in a large drop of reaction mixture (KSOM containing 1/250 Magic Red stock solution dissolved in DMSO and 1/1000 LysoSensor Green DND-189 stock solution dissolved in DMSO) for 25 min at 37°C under 5% $CO_2$ in a humidified atmosphere in liquid paraffin in a 35 mm glass-bottomed dish (MatTek Corporation). Thereafter, Hoechst 33342 (10 μg/mL) was added to the drop to a final concentration of 10 μg/mL, and incubated for 5 min. Subsequently, oocytes or embryos at the same stage were collected, and confocal images at 41 z-axis planes with 2 μm increments for both 470 nm (LysoSensor) and 555 nm (Magic Red) lasers were obtained on a CSU W-1 spinning disk confocal microscopy using a 30× silicone immersion lens UPLSAPO30XS (Olympus Corporation).

## In-resin CLEM and electron microscopy

In-resin CLEM was performed as previously described with some modifications (*Tanida et al., 2023*; *Mitsui et al., 2023*). Briefly, oocytes were fixed in 4% PFA and 0.025% GA. After permeabilization with 50 μg/mL digitonin (Nacalai Tesque) and 0.02% Triton X-100, oocytes were washed with PVA/PBS three times and incubated with PBS containing 10% (wt/vol) FBS (Thermo Fisher Scientific) for 1 hr at room temperature. Thereafter, oocytes were incubated with PVA/PBS containing primary antibody overnight at 4°C. After incubation with a secondary antibody in PVA/PBS for 2 hr, oocytes were fixed in 4% PFA and 2.5% GA, embedded in 1.5% agarose, stained with osmium tetroxide, dehydrated with graded ethanol solutions (QY1), and embedded in epoxy resin (Oken Shoji, Tokyo, Japan). Ultrathin sections of Epon-embedded oocytes (100 nm thickness) were prepared using a Leica UC6 ultramicrotome (Leica, Nussloch, Germany) and placed on glass coverslips. Subsequently, fluorescence microscopy images of the sections were obtained using a Nikon A1RHD25 confocal laser scanning microscope with a NIS-Elements software (Nikon, Tokyo, Japan). Electron microscopic images were observed using a Helios Nanolab 660 FIB-SEM instrument (FEI Company, Hillsboro, OR, USA).

For conventional electron microscopy, the oocytes were fixed in the presence of 2% PFA and 2.5% GA and stained with osmium tetroxide. After dehydration with ethanol, the oocytes were embedded in epoxy resin, as described above. Sections were cut at a thickness of 100 nm using a UC6 ultramicrotome, mounted on glass coverslips, stained with uranyl acetate and lead citrate, and observed under a Regulus8240 scanning electron microscope (Hitachi High-Tech Corporation, Tokyo, Japan).

## Serial semi-thin sections and 3D reconstruction of toluidine blue-positive structures

Serial semi-thin sections were cut at a thickness of 300 nm using a UC6 ultramicrotome and mounted on glass slides. Next, the sections were stained with 1% toluidine blue/0.1 M PB, and images were obtained using a BX50 light microscope (Olympus Corporation). Serial toluidine blue-stained images were aligned using FIJI (National Institutes of Health, MD, USA) with plugin-linear stack alignment using SIFT (LSAS), and 3D reconstruction was performed using Amira software (Thermo Fisher Scientific).

## Microinjection into the oocytes

Microinjection of mRNA into GV oocytes was performed as previously described (*Umeda et al., 2020*; *Kato et al., 2016*). Briefly, antral follicles were harvested in FHM medium containing 250 µM of dbcAMP using 26G needles (Terumo). After dissociating the cumulus cells by pipetting, the oocytes were transferred to medium supplemented with 250 µM of dbcAMP and 10 µM of CCB (Sigma-Aldrich). Thereafter, mRNA injection into the oocytes was performed using a piezo-micromanipulator (Primetech, Tokyo, Japan) with a glass capillary needle. The mRNA concentrations were 50 and 100 ng/µL for *V1A-mCherry* and *Lamp1-EGFP*, respectively. The oocytes were washed three times and used for IVM. Microinjection of mRNA into zygote stage embryo (at 3–4 hpf) was carried out as previously described (*Morita et al., 2021*). The concentration of mRNA was the same as that in the GV oocytes.

## Time-lapse confocal imaging of oocytes/embryos

Low-invasive confocal fluorescence imaging of oocytes/embryos was performed as previously described (*Nozawa et al., 2018*; *Satouh et al., 2017*; *Satouh et al., 2012*) with slight modifications. Briefly, on an inverted microscope IX-71 (Olympus Corporation) equipped with a CSU W-1 confocal scanner unit (Yokogawa Electric Corporation), an incubation chamber (Tokai Hit, Fujinomiya, Japan) was set at 37°C on the microscope stage, with a gas mixture of 5% $CO_2$ introduced into the chamber at 150 mL/min. Oocytes/embryos were placed in 8 µL of IVM medium or KSOM drops covered with liquid paraffin (Nacalai Tesque) in 35 mm glass-bottomed dishes (MatTek Corporation). Subsequently, confocal images at 41 z-axis planes with 2 µm increments for excitation lasers of 470 and 555 nm from an LDI-7 Laser Diode Illuminator (89 North, Williston, VT, USA) were obtained every 30 or 60 min during IVM or embryonic culture observation, respectively, through optimized band-pass filters using 30× silicone immersion lens UPLSAPO30XS (Olympus Corporation).

## 3D object analysis

Series of confocal images with 81 z-axis planes with 1 µm increments for LAMP1 immunostaining were used for 3D object analysis. Fields to analyze were cropped for each oocyte/embryo, and the upper and lower end of LAMP1 fluorescent signals in the cytosol were examined to determine the center plane (middle height) of the cell. The cellular region in the center plane was used for masking and determination of the geometric center (as a 3D center) of each oocyte. Deletion of the signals from the polar body or zona pellucida was carried out manually. Objects were profiled using the 3D object counter plugin (*Bolte and Cordelières, 2006*) of Fiji software (*Schindelin et al., 2012*), to measure the number, volume, geometric center of each object, and mean distance from the geometrical center of the object to its surface. Thresholding was carried out after background subtraction using the Fiji software, manually applying intensities of 600–1200 as threshold to recognize objects separately. Signals on PM were eliminated both via size filtering (0–5000 µm³) of the 3D object counter plugin and manual erasing. The distal distance of each object from the 3D center of the oocyte was used for distribution analysis to avoid underestimation of the distance of large ELYSA. This was calculated by adding the distance from the 3D center of the oocyte to the geometric center of each object and the mean distance from the geometrical center of the object to its surface. Distance calculation and generation of bar graphs were carried out using Excel 2016 software (Microsoft, Redmond, WA, USA). Object-based colocalization analysis for the punctate structures were performed using the JACoP plugin in the ImageJ Fiji software (*Bolte and Cordelières, 2006*). Confocal z-stack images of multiple wave length for LAMP1 and LC3 or RAB5 antibodies were acquired for each oocyte as above, and object-based colocalization was examined by identifying coincidence of geometric centers of LC3 or RAB5 objects overlapping on LAMP1 objects. All of the LAMP1-positive 3D objects were listed with the 3D

object counter plugin applied with same threshold, and the colocalized LAMP1-positive objects were identified and analyzed using the Excel 2016 software.

## Analysis of microscopic images and movies

Analysis of the acquired images, including the generation of a line plot of intensity, adjustment of intensity, and generation of montage images at a uniform intensity, was performed using Fiji software (*Schindelin et al., 2012*). To enhance the images, a Gaussian Blur filter (at a sigma radius of 1) and auto-adjustment of the intensity were applied to the stack of images. To adjust the intensity, auto-adjustment of the intensity was applied to each image. The movies were concatenated using Premiere Pro software (Adobe, San Jose, CA, USA).

## Statistics and reproducibility

Sample sizes were estimated based on previous studies using similar experiments, and results from preliminary experiments were examined to ensure statistical power more than 0.8 using statistical power analysis in the G*Power software (*Faul et al., 2007*). Statistical analyses and generation of dot plot graphs were performed using GraphPad Prism 8 (GraphPad Software, Boston, MA, USA). The significance of difference among groups were compared using one-way ANOVA and Tukey's multiple comparison test. The experimenters were not blinded because of a limitation in the availability of experienced personnel. Images indicated in the figures were selected as representative among at least 10 oocytes/embryos from independent experiments of three times or more.

## Materials availability statement

The vector for the expression of *Lamp1-EGFP* or *Atp6v1a-mCherry* is available from the corresponding authors upon reasonable request.

# Acknowledgements

We would like to thank Editage (https://www.editage.jp/) for English language editing.

This study was supported by the Japan Society for the Promotion of Science KAKENHI (Grant Numbers 19H05711, 20H00466, 24H02031 to K Sato, 21H02435, 22H02872, and 22H04652 to I Tanida, and 20K22744 to J Yamaguchi), Takeda Science Foundation to K Sato (Grant Number 2024095741), the Mochida Memorial Foundation for Medical and Pharmaceutical Research, Takeda Science Foundation (Grant Number 2022093867), and The Cell Science Research Foundation to Y Satouh. This work was partly supported by the Project for Elucidating and Controlling Mechanisms of Aging and Longevity from the Japan Agency for Medical Research and Development (AMED 21gm5010003 to Y Uchiyama, 22gm1710001 s0201 and 23gm1710001s0202 to I Tanida) and by the MEXT-supported Program for the Strategic Research Foundation at Private Universities (to Y Uchiyama), and the Center of Genomic and Regeneration Medicine, Juntendo University Graduate School of Medicine (to Y Uchiyama and I Tanida).This work was partly supported by the joint research program of the Institute for Molecular and Cellular Regulation, Gunma University (24014 to I Tanida and K Sato).

# Additional information

## Funding

| Funder | Grant reference number | Author |
|--------|------------------------|--------|
| Japan Society for the Promotion of Science | 19H05711 | Ken Sato |
| Japan Society for the Promotion of Science | 20H00466 | Ken Sato |
| Japan Society for the Promotion of Science | 24H02031 | Ken Sato |
| Japan Society for the Promotion of Science | 21H02435 | Isei Tanida |

| Funder | Grant reference number | Author |
|---|---|---|
| Japan Society for the Promotion of Science | 22H02872 | Isei Tanida |
| Japan Society for the Promotion of Science | 22H04652 | Isei Tanida |
| Japan Society for the Promotion of Science | 20K22744 | Junji Yamaguchi |
| Takeda Science Foundation | 2024095741 | Ken Sato |
| Mochida Memorial Foundation for Medical and Pharmaceutical Research | | Yuhkoh Satouh |
| Cell Science Research Foundation | | Yuhkoh Satouh |
| Takeda Science Foundation | 2022093867 | Yuhkoh Satouh |
| Japan Agency for Medical Research and Development | 21gm5010003 | Yasuo Uchiyama |
| Japan Agency for Medical Research and Development | 22gm1710001s0201 | Isei Tanida |
| Japan Agency for Medical Research and Development | 23gm1710001s0202 | Isei Tanida |
| Ministry of Education, Culture, Sports, Science and Technology | | Yasuo Uchiyama |
| Juntendo University | | Isei Tanida Yasuo Uchiyama |
| Institute for Molecular and Cellular Regulation, Gunma University | 24014 | Isei Tanida Ken Sato |

The funders had no role in study design, data collection and interpretation, or the decision to submit the work for publication.

## Author contributions

Yuhkoh Satouh, Conceptualization, Data curation, Formal analysis, Supervision, Funding acquisition, Validation, Investigation, Visualization, Methodology, Writing – original draft, Project administration, Writing – review and editing; Takaki Tatebe, Resources, Investigation, Visualization; Isei Tanida, Data curation, Funding acquisition, Validation, Investigation; Junji Yamaguchi, Data curation, Funding acquisition, Validation, Investigation, Visualization; Yasuo Uchiyama, Data curation, Supervision, Funding acquisition; Ken Sato, Conceptualization, Supervision, Funding acquisition, Writing – original draft, Writing – review and editing

## Author ORCIDs

Yuhkoh Satouh ⓘ https://orcid.org/0000-0001-9207-9374
Takaki Tatebe ⓘ http://orcid.org/0009-0003-8998-3567
Isei Tanida ⓘ http://orcid.org/0000-0001-8999-3990
Junji Yamaguchi ⓘ http://orcid.org/0009-0002-9199-9931
Yasuo Uchiyama ⓘ http://orcid.org/0000-0002-9104-533X
Ken Sato ⓘ https://orcid.org/0000-0002-1034-5091

## Ethics

All experimental protocols involving animals were approved and performed in accordance with the guidelines of the Animal Care and Experimentation Committee of Gunma University (Approval No. 22-076).

Reviewer #1 (Public review): https://doi.org/10.7554/eLife.99358.3.sa1
Reviewer #2 (Public review): https://doi.org/10.7554/eLife.99358.3.sa2
Author response https://doi.org/10.7554/eLife.99358.3.sa3

---

# Additional files

## Supplementary files

MDAR checklist

## Data availability

Figure 2-Source Data 1, Figure 2-figure supplement 2-Source Data 1, Figure 4-figure supplement 1-Source Data 1, Figure 5-Source Data 1, Figure 6-figure supplement 1-Source Data 1, and Figure 9-Source Data 1 contain the numerical data used to generate the figures.

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
