## [Editor Report · eLife Assessment]

This paper reports **important** findings on giant organelle complexes containing endosomes and lysosomes (termed endosomal-lysosomal organelles form assembly structures [ELYSAs]) present in mouse oocytes and 1- to 2-cell embryos. The data showing the localization and dynamics of ELYSAs during oocyte/embryo maturation are **convincing**. This work will be of interest to general cell biologists and developmental biologists.

---

## [Referee Report · Reviewer #1 (Public review)]

Satouh et al. report giant organelle complexes in oocytes and early embryos. Although these structures have often been observed in oocytes and early embryos, their exact nature has not been characterized. The authors named these structures "endosomal-lysosomal organelles form assembly structures (ELYSAs)". ELYSAs contain organelles such as endosomes, lysosomes, and probably autophagic structures. ELYSAs are initially formed in the perinuclear region and then seem to migrate to the periphery in an actin-dependent manner. When ELYSAs are disassembled after the 2-cell stage, the V-ATPase V1 subunit is recruited to make lysosomes more acidic and active. The ELYSAs are most likely the same as the "endolysosomal vesicular assemblies (ELVAs)", reported by Elvan Böke's group earlier this year (Zaffagnini et al. doi.org/10.1016/j.cell.2024.01.031). However, it is clear that Satouh et al. identified and characterized these structures independently. These two studies could be complementary. Although the nature of the present study is generally descriptive, this paper provides valuable information about these giant structures. Since the ELYSA described in this paper and ELVA proposed by Elvan Böke appear to be the same structure, it would be helpful to the field if the two groups discuss unifying the nomenclature in the future.

Comments on latest version:

In this revised manuscript, the authors have provided additional data supporting their conclusions and also revised the text to more accurately reflect the experimental results.

---

## [Referee Report · Reviewer #2 (Public review)]

Satouh et al report the presence of spherical structures composed of endosomes, lysosomes and autophagosomes within immature mouse oocytes. These endolysosomal compartments have been named as Endosomal-LYSosomal organellar Assembly (ELYSA). ELYSAs increase in size as the oocytes undergo maturation. ELYSAs are distributed throughout the oocyte cytoplasm of GV stage immature oocytes but these structures become mostly cortical in the mature oocytes. Interestingly, they tend to avoid the region which contain metaphase II spindle and chromosomes. They show that the endolysosomal compartments in oocytes are less acidic and therefore non-degradative but their pH decreases and become degradative as the ELYSAs begin to disassemble in the embryos post fertilization. This manuscript shows that lysosomal switching does not happen during oocyte development, and the formation of ELYSAs prevent lysosomes from being activated. Structures similar to these ELYSAs have been previously described in mouse oocytes (Zaffagnini et al, 2024) and these vesicular assemblies are important for sequestering protein aggregates in the oocytes but facilitate proteolysis after fertilization. The current manuscript, however, provides further details of endolysosomal disassembly post fertilization. Specifically, the V1-subunit of V-ATPase targeting to the ELYSAs increases the acidity of lysosomal compartments in the embryos. This is a well-conducted study and their model is supported by experimental evidence and data analyses.

Comments on revisions:

This revised version of the manuscript has addressed most of my concerns.

---

## [Author Response]

The following is the authors’ response to the original reviews.

**Public Reviews:**

**Reviewer #1 (Public Review):**
In this manuscript, Satouh et al. report giant organelle complexes in oocytes and early embryos. Although these structures have often been observed in oocytes and early embryos, their exact nature has not been characterized. The authors named these structures "endosomal-lysosomal organelles form assembly structures (ELYSAs)". ELYSAs contain organelles such as endosomes, lysosomes, and probably autophagic structures. ELYSAs are initially formed in the perinuclear region and then migrate to the periphery in an actin-dependent manner. When ELYSAs are disassembled after the 2-cell stage, the V-ATPase V1 subunit is recruited to make lysosomes more acidic and active. The ELYSAs are most likely the same as the "endolysosomal vesicular assemblies (ELVAs)", reported by Elvan Böke's group earlier this year (Zaffagnini et al. doi.org/10.1016/j.cell.2024.01.031). However, it is clear that Satouh et al. identified and characterized these structures independently. These two studies could be complementary. Although the nature of the present study is generally descriptive, this paper provides valuable information about these giant structures. The data are mostly convincing, and only some minor modifications are needed for clarification and further explanation to fully understand the results.
**Reviewer #2 (Public Review):**
Satouh et al report the presence of spherical structures composed of endosomes, lysosomes, and autophagosomes within immature mouse oocytes. These endolysosomal compartments have been named as Endosomal-LYSosomal organellar Assembly (ELYSA). ELYSAs increase in size as the oocytes undergo maturation. ELYSAs are distributed throughout the oocyte cytoplasm of GV stage immature oocytes but these structures become mostly cortical in the mature oocytes. Interestingly, they tend to avoid the region which contains metaphase II spindle and chromosomes. They show that the endolysosomal compartments in oocytes are less acidic and therefore non-degradative but their pH decreases and becomes degradative as the ELYSAs begin to disassemble in the embryos post-fertilization. This manuscript shows that lysosomal switching does not happen during oocyte development, and the formation of ELYSAs prevents lysosomes from being activated. Structures similar to these ELYSAs have been previously described in mouse oocytes (Zaffagnini et al, 2024) and these vesicular assemblies are important for sequestering protein aggregates in the oocytes but facilitate proteolysis after fertilization. The current manuscript, however, provides further details of endolysosomal disassembly post-fertilization. Specifically, the V1-subunit of V-ATPase targeting the ELYSAs increases the acidity of lysosomal compartments in the embryos. This is a well-conducted study and their model is supported by experimental evidence and data analyses.
**Reviewer #3 (Public Review):**
Fertilization converts a cell defined as an egg to a cell defined as an embryo. An essential component of this switch in cell fate is the degradation (autophagy) of cellular elements that serve a function in the development of the egg but could impede the development of the embryo. Here, the authors have focused on the behavior during the egg-to-embryo transition of endosomes and lysosomes, which are cytoplasmic structures that mediate autophagy. By carefully mapping and tracking the intracellular location of well-established marker proteins, the authors show that in oocytes endosomes and lysosomes aggregate into giant structures that they term Endosomal LYSosomal organellar Assembl[ies] (ELYSA). Both the size distribution of the ELYSAs and their position within the cell change during oocyte meiotic maturation and after fertilization. Notably, during maturation, there is a net actin-dependent movement towards the periphery of the oocyte. By the late 2-cell stage, the ELYSAs are beginning to disintegrate. At this stage, the endo-lysosomes become acidified, likely reflecting the activation of their function to degrade cellular components.This is a carefully performed and quantified study. The fluorescent images obtained using well-known markers, using both antibodies and tagged proteins, support the interpretations, and the quantification method is sophisticated and clearly explained. Notably, this type of quantification of confocal z-stack images is rarely performed and so represents a real strength of the study. It provides sound support for the conclusions regarding changes in the size and position of the ELYSAs. Another strength is the use of multiple markers, including those that indicate the activity state of the endo-lysosomes. Altogether, the manuscript provides convincing evidence for the existence of ELYSAs and also for regulated changes in their location and properties during oocyte maturation and the first few embryonic cell cycles following fertilization.At present, precisely how the changes in the location and properties of the ELYSAs affect the function of the endo-lysosomal system is not known. While the authors' proposal that they are stored in an inactive state is plausible, it remains speculative. Nonetheless, this study lays the foundation for future work to address this question.Minor point: l. 299. If I am not mistaken, there is a typo. It should read that the inhibitors of actin polymerization prevent redistribution from the cytoplasm to the cortex during maturation.Minor point: A few statements in the Introduction would benefit from clarification. These are noted in the comments to the authors.

We sincerely appreciate the editorial board of *eLife* and the reviewers for their helpful and constructive comments on our manuscript. We are pleased that the reviewers acknowledged that we identified and characterized this assembly structure independently. In the revised manuscript, we have carefully considered the reviewers’ comments and conducted additional analysis to address each of them.

Regarding the typographical errors, we revised the description to fit with our findings and the reviewers’ comments. We also found that the primer sequence was correct, and we carefully checked the accuracy of the entire manuscript.

We hope that the revised version will now be deemed suitable for publication in *eLife*.

**Recommendations for the authors:**

**Reviewer #1 (Recommendations For The Authors):**
Q. (1) The authors state in the Abstract that ELYSAs contain autophagosome-like membranes in the outer layer. However, this seems to be just speculation based on the LC3 staining results and is not directly shown. Are there autophagosome-like double membrane structures in ELYSAs?

We appreciate this comment. We also agree with this concern; however, it was difficult to assert that they are autophagosomes based on the observation of the electron micrographs. For this reason, we rephrased it to be "Most ELYSAs are also positive for an autophagy regulator, LC3.” (lines 33). In addition, we revised the notation to LC3-positive structures in the Result and Discussion section (line 165-169, 286).

Q. (2) The data in Figure 2A, showing a decrease in the number of LAMP1 structures, seems to contradict the data in Figure 1B, showing an apparent increase in LAMP1 structures. Please explain this discrepancy. If the authors did not count structures just below the plasma membrane, please explain the rationale for this.

We really appreciate the valuable comment. Regarding the number of LAMP1-positive structures, it is not suitable for comparison with Figure 1B, etc., as pointed out by the reviewer, since the distribution of the LAMP1 signal differs from plane to plane. To avoid any potential confusion, we added new images of the Z-projection of the immunostained images that can better reflect the number of positive structures in the whole oocyte/embryo in Figure 2.

In addition, as the reviewer pointed out, there is a technical difficulty in measuring the LAMP1-positive signal on the plasma membrane or just below it. We explained how and why we had to delete plasma membrane signals in our response #21.

Q. (3) The actin dependence is not observed in Figure 5C. What is the difference between Figure 5C and 5E? Please explain further.

We apologize for the lack of clarity; Figures 5C and 5E show the average number of LAMP1-positive structures (5C) and the percentage of the sum of granule volumes in LAMP1 positive structure (5E), respectively, after classifying the LAMP1 positive granules by their diameters.

We removed Figure 5E for the sake of conciseness since we already mentioned a similar fact in Figure 5C. To clarify the corresponding explanations, we moved figures that were not classified by diameter to Supplementary Figure 8 to improve readability. Moreover, we have rewritten the main text on lines 200–211.

Q. (4) While the actin inhibitors reduce the number of peripheral LAMP1 structures (Figure 5F), they do not affect their number in the central region (Figure 5G). How can the authors conclude that actin inhibitors inhibit the migration of LAMP1 structures?

We appreciate the comment. As pointed out, the number of large LAMP1-positive structures in the medial region did not change. Therefore, we have avoided the description that ELYSAs migrate from the middle region to the cell periphery and have unified the description of whether large structures in the periphery occur. Please refer to the subsection title (line 188), the following descriptions (lines 189–199), the related description in the Results (lines 200–211), and the title and the legend of Figure 5.

Q. (5) The authors show that the V1A subunit associates with the surface of LAMP1 structures as punctate structures (Figure 6B). What are these V1A-positive structures? Is V1A recruited to some specific domains of ELYSAs, or are V1A-positive active lysosomes recruited to ELYSAs? Please provide an interpretation of these data. The phrase "The V1-subunit of V-ATPase is targeted to these structures" (line 262) is not appropriate because it is indistinguishable whether only the V1 subunits are recruited or active lysosomes containing the V1 subunit are recruited.

Thank you for the valuable comment. Indeed, our analysis, including the analysis of Fig. 8 described on line 262, did not clarify whether free V1A-mCherry molecules accessed the ELYSA periphery or whether lysosomes with V1A-mCherry molecules newly merged into the ELYSA. Therefore, we added this interpretation to lines 232–234 of the Results and revised the Discussion as "The number of membrane structures positive for V1A-mCherry increase upon ELYSA disassembly, indicating further acidification of the endosomal/lysosomal compartment" (lines 292–294).

Q. (6) Why did the authors use LysoSensor as a marker for ELYSA instead of LAMP1 in Figure 8 and 9? Some reasons should be given.

There is a clear technical reason for this: when LAMP1-EGFP was expressed in a zygote, it was largely migrated to the plasma membrane before and after the 2-cell stage, making it difficult to capture the change of ELYSAs. To circumvent this difficulty, we used Lysosensor to visualize ELYSAs instead of LAMP1-EGFP. This explanation was added to lines 258–260.

Q. (7) In Figure 9A, it is not clear whether the activity of LysoSensor-positive structures is lower at this stage compared to other stages. It may be shown in Figure S7, but the data are not clearly visible. A direct comparison would be ideal.

A new analysis similar to that shown in Fig. 9 for early 2-cells and 4-cells was performed and added to Figure S7. To support direct comparison, the ranges of axes were set to be similar.

As a result, the quantified MagicRed signal on the isolated LysoSensor-positive punctate structure in MII oocyte was nearly the same as that in early 2-cells and 4-cells. In early 2-cells, LysoSensor gave a signal at the cellular boundary, where MagicRed staining was not observed, confirming that MagicRed activity is higher in the interior than in the cell periphery in post-fertilization embryos. We have included an additional description in the main text (lines 280–282).

Q. (8) In the phrase "pregnant mare serum gonadotropin or an anti-inhibin antibody" (line 382), is "or" correct?

When inducing superovulatory stimulation, an anti-inhibin antibody (distributed as CARD HyperOva) can be used as a substitute for PMSG (after additional stimulation with hCG), which results in the production of eggs of similar quality to those of PMSG. This was used in most experiments. To amend the lack of clarity, a reference (Takeo and Nakagata Plos One, 2015) was added to the description of HyperOva (line 417).

Q. (9) In almost all graphs, please indicate what the X-axis is indicating (not just "number") so that readers can understand what number is being represented without reading the legends.

We revised the axis titles in all figures.

Q. (10) Since grayscale images provide better contrast than color images, it is recommended that single-color images be shown in grayscale.

We replaced all single-color images with grayscale images.

**Reviewer #2 (Recommendations For The Authors):**
Specific comments:Q. (11) Figure 1 and S1- Both Rab5 and Rab7 co-localize with LAMP1. However, there seems to be a lot of LAMP1-free Rab5 dots as compared to the Lamp1-free Rab7. As a result, LAMP1 and Rab7 are co-localized more frequently than LAMP1 and Rab5 (video1). Could it be that early endosomes (Rab5+) are yet to be incorporated into ELYSAs? If so, a brief discussion of this phenomenon would be nice.

Thank you very much for the comment. We agree with the reviewer’s interpretation. In accordance with this suggestion, we clearly stated in the main text: “Although small punctate structures that are RAB5-positive but LAMP1-negative also spread over the cytosol, most giant structures were positive for RAB5 and LAMP1 (Video 1)” (lines 91–93). In the Discussion section, a brief statement was included: “Considering the large number of RAB5-positive and LAMP1-negative punctate structures in MII oocytes, these layers may also reflect the assembly mechanism of the ELYSA” (lines 318–320).

Q. (12) Video 3 (and Figure 6) clearly shows the dynamics of LAMP1-labelled vesicles during maturation, which is impressive. In contrast to the live cell imaging after LAMP1 mRNA injection, Figure 1 used anti-LAMP1 Ab to detect endogenous levels of LAMP1. It appears that mRNA microinjection causes LAMP1 overexpression causing more (but smaller) vesicles to form. It should be easy to quantify and compare the vesicles in Figure 1 and 6

We appreciate the comment. As mentioned, injections of EGFP-LAMP1 mRNA are useful for the visualization of LAMP1 dynamics during the maturation phase from GV to MII by live cell imaging, which is not feasible with immunostaining. However, the fluorescence emitted by EGFP-LAMP1 is only a few tenths of that of antibody staining, and because of the technical difficulty of microinjection into GV oocytes, the signal-to-noise ratio sufficient for imaging was merely one in ten oocytes. In addition, live cell imaging of oocytes in Figure 6 had to be carried out with very low excitation light exposure to reduce the toxicity. It was also performed with a low magnification lens and a longer step size in the z-axis. For these reasons, in examining the point raised, we performed an additional 3D object analysis, in the same way as in Figure 2, on the data of IVM oocytes injected with EGFP-LAMP1 mRNA using the same lens as in Figure 1 and with a longer exposure time than in live imaging. The results were compared with the MII data of Figures 1 and 2.

As a result, as shown in the new Figure S8, more objects with a diameter of 0.2–0.4 µm were found than in the immunostaining data, which fits the reviewer’s point. In addition, the counts were lower for the 0.6–1.0 µm diameter, but there was no significant difference in the number of larger LAMP1 positive structures corresponding to the ELYSA size. We consider that this was appropriate for the original purpose of characterizing the ELYSA formation process. A description of these points has been added to lines 221–225.

Q. (13) In Figure 4A and B- Seems like not all LAMP1-positive structures were LC3-positive. Is there any size or location within the oocyte that determines LC3 positivity?

We appreciate the valuable comment. To answer this comment, we proceeded with a new 3D object-based co-localization analysis on Lamp1 and LC3, determined the number, volume, and distribution within the oocyte, and incorporated the results as Supplementary Figure 6. To examine the positivity, we further analyzed the percentage of double-positive structures of all the LAMP1-positive structures. The results showed that their average diameter significantly shifted from 2.36 µm (GV) to 3.78 µm (MII). Moreover, it was clearly indicated that LAMP1-positive structures smaller than 2 µm in diameter are rarely positive for LC3. In terms of location, measuring the distance of the double positive structures from the oocyte center (the cellular geometric center) indicated that they tend to be observed at the periphery of both stages of oocytes (more than 80% in > 30 µm in the MII oocyte). Of note, no clear tendency of double positivity was observed. A description of these points has been added to lines 174–186.

Q. (14) In discussion, line 256- Small ELYSAs are formed in GV oocytes. Since you haven't checked the smaller-sized, growing oocytes, I suggest rephrasing this sentence as 'are present' rather than 'are formed'.

We agree with the reviewer’s suggestion and changed it to "present" (line 287).

Q. (15) Line 188- ELISA should instead be ELYSA

Thank you for pointing this out. We have found a few more typographical errors, and all of them have been corrected (lines 213 and 321).

**Reviewer #3 (Recommendations For The Authors):**
Q. (16) Line 42: What do you mean by 'zygotic gene expression following the degradation of the cellular components of each maternal and paternal gamete'? ZGA requires this degradation? Please provide supporting references from the literature.

We apologize for the confusing wording. We meant to say that both ZGA and degradation of parental components are required. To avoid misunderstanding, we have revised “zygotic gene expression as well as the degradation of the cellular components of each maternal and paternal gamete” and inserted a new reference (line 44).

Q. (17) 50: MII means metaphase II, not meiosis II.

We corrected the clerical mistake (line 50).

Q. (18) 51: Define LC3.

We added the definition of LC3 (line 51-52).

Q. (19) 60: 'lysosomal activity in oocytes is upregulated by sperm-derived factors as the oocytes grow and mature'. As written, the sentence implies that oocytes grow and mature after fertilization. This may be true for maturation, but I would be surprised to learn that there is growth of the oocyte after fertilization.

We appreciate this valuable comment.

The *C. elegans* lives mainly as a hermaphrodite, which contains a couple of U-shaped gonad arms including the ovary, spermatheca and uterus in the body. Oocytes grow in the ovary and maturate upon receiving major sperm proteins secreted from sperms and ovulated to the spermatheca for fertilization. In 2017, Kenyon’s group reported that major sperm proteins act as sperm-secreted hormones to upregulates the lysosomal activity in oocytes during oocyte growth and maturation. We have revised our manuscript to avoid misunderstanding, to ' lysosomal activity in oocytes is upregulated by major sperm proteins secreted from sperms as the oocytes grow and mature '. (L. 61-66).

Q. (20) 94 and Figure 1B: While it is clear that many LAMP1 foci at the late 2-cell stage do not also contain RAB5, it seems that the majority of RAB5 loci also stain for LAMP1. This may be a minor point in the context of the paper but could be clarified.

We could not easily agree with the suggestion because of the possibility that the images might give different impressions on each plane. Therefore, as a way to verify this point, we attempted to quantify the co-localization by reconstructing the 3D puncta information based on the two types of antibody staining data. Unfortunately, as shown in Fig. 1AB, Rab5 had a high cytoplasmic background, and although we were able to extract peaks, we could not reliably recalibrate the three-dimensional punctate structure (please refer to the new Supplementary Fig. 6). Therefore, co-localization on each other's punctate structure (LAMP1/RAB5 vs. RAB5/LAMP1) could not be verified. The validation using specific planes also showed large differences between planes, with overlapping punctate structures counted separately in adjacent planes, making reliable quantification difficult. This is an issue that will be addressed in the future.

On the other hand, the newly added Z-projection figure (Fig. 1AB) shows that RAB5-positive and LAMP1-negative punctate structures tend to accumulate along the LAMP1-positive punctate structures larger than 1 µm at the late 2-cell stage in all observed embryos; we added this statement on lines 99–101.

Q. (21) 100-102 and Figure 2A: Does the decrease in the total number of LAMP1 foci refer just to cytoplasmic or also to membrane foci? If the former, what was the reason for not including the membrane in the analysis?

We appreciate the critical question. The LAMP1 signal on the plasma membrane interfered with the measurement of the signals just below the plasma membrane. The biological cause of this increased signal on the plasma membrane, as shown in Fig. 2E, seemed to be caused by the migration of the LAMP1 signals post-fertilization, which was also reported in a previous paper by Zaffagnini et al. (2024), published in *Cell*.

In our analysis, oocytes are giant cells, and confocal imaging has a technical limitation in obtaining the same fluorescent intensity along the z-axis. However, 3D-object analysis requires thresholding based on absolute values. As a result of this situation, the presence of the plasma membrane signal caused punctate structures located close to the membrane to be captured and recognized as a single, very large LAMP1-positive structure, resulting in the loss of the punctate structure that should be measured.

To avoid this issue, we have used several programs to correct the fluorescence difference along the z-axis; nonetheless, these attempts were unsuccessful. Therefore, as described in the Materials and Methods section, we applied only background subtraction at each z-position and then manually removed the plasma membrane signal (which was thin and continuous at the edges). Furthermore, when the plasma membrane and punctate structure signals overlapped, we paid attention not to remove the signals but to separate them. Thus, we believe that the decrease in the number and volume of LAMP1-positive structures after fertilization is still a phenomenon associated with the shift of LAMP1 to the plasma membrane.

Q. (22) Figure 2B, F, G: As the x-axis does not represent a continuous variable, adjacent data points should not be connected by a line. The histogram representations in A, C, and E are much easier to understand. I suggest presenting all data in this format.

We revised the line graphs to bar graphs. Besides, to make the significance among populations clearer, the significances are now expressed using alphabetical indicators.

Q. (23) Figure 2B, C: It seems that the values for the different stages are expressed relative to the value at MII. Why not use the GV value at the base-line? This would follow the developmental trajectory of the oocyte/embryo more directly and would not (I believe) change the conclusions.

We appreciated the comment. We meant to express that ELYSA develops most in the MII phase and that it decreases after fertilization, so considering the reviewer’s suggestion, we expressed GV-MII changes based on GV and changes after fertilization based on the MII phase (Fig. 2C, D).